# Granger causality analysis for calcium transients in neuronal networks, challenges and improvements

**Xiaowen Chen[1†], Faustine Ginoux[2†], Martin Carbo-Tano[2], Thierry Mora[1*‡], Aleksandra M Walczak[1*‡], Claire Wyart[2*‡]**

[1]Laboratoire de physique de l'École normale supérieure, CNRS, PSL University, Paris, France; [2]Spinal Sensory Signaling team, Sorbonne Université, Paris Brain Institute (Institut du Cerveau, ICM), Paris, France

**Abstract** One challenge in neuroscience is to understand how information flows between neurons *in vivo* to trigger specific behaviors. Granger causality (GC) has been proposed as a simple and effective measure for identifying dynamical interactions. At single-cell resolution however, GC analysis is rarely used compared to directionless correlation analysis. Here, we study the applicability of GC analysis for calcium imaging data in diverse contexts. We first show that despite underlying linearity assumptions, GC analysis successfully retrieves non-linear interactions in a synthetic network simulating intracellular calcium fluctuations of spiking neurons. We highlight the potential pitfalls of applying GC analysis on real *in vivo* calcium signals, and offer solutions regarding the choice of GC analysis parameters. We took advantage of calcium imaging datasets from motoneurons in embryonic zebrafish to show how the improved GC can retrieve true underlying information flow. Applied to the network of brainstem neurons of larval zebrafish, our pipeline reveals strong driver neurons in the locus of the mesencephalic locomotor region (MLR), driving target neurons matching expectations from anatomical and physiological studies. Altogether, this practical toolbox can be applied on *in vivo* population calcium signals to increase the selectivity of GC to infer flow of information across neurons.

**\*For correspondence:**
thierry.mora@phys.ens.fr (TM);
aleksandra.walczak@phys.ens.fr (AMW);
claire.wyart@icm-institute.org (CW)

†These authors contributed equally to this work
‡These authors also contributed equally to this work

## Editor's evaluation

This important paper provides an in-depth analysis of the advantages and potential pitfalls of the application of Granger Causality (GC) to calcium imaging data, especially regarding various types of pre-processing. The authors' approach uses compelling rigor in arguing their points, and it is very clear how one would go about replicating their work. These results should be of interest to any researcher attempting to analyze calcium imaging data.

## Introduction

The prompt integration of sensory inputs by motor command neurons in the nervous systems can become a matter of life and death, often requiring fast and precise movements in response to the stimulus. Understanding how large neuronal networks coordinate to generate cognitive activities requires knowing how information from the stimuli flows between neurons to result in the observed behavior. Recent developments in optogenetics (***Deisseroth, 2011***; ***Fenno et al., 2011***), and whole-brain imaging with single-cell resolution (***Ahrens et al., 2013***; ***Nguyen et al., 2016***; ***Venkatachalam et al., 2016***; ***Cong et al., 2017***) allow us to observe collective firing patterns of many neurons and give the basis for reconstructing a causality network that describes how neurons influence each other.

However, despite the existing data, assessing the direction of neuronal communication or causality in order to understand how information flows in the brain during integrative reflexive behaviors remains a major challenge in neuroscience (*Goulding, 2009*).

Apart from perturbing individual neurons and observing the activity of downstream neurons, non-invasive methods for obtaining functional connectivity have been developed (*Bastos and Schoffelen, 2016*). The widely-used correlation analysis (*Cohen and Kohn, 2011*; *Haesemeyer et al., 2018*; *Stringer et al., 2019*) identifies pairs or groups of neurons that are active at the same time, suggesting either that they drive each other, or are driven by a common input. By contrast, Granger causality (GC) identifies directed interactions (*Granger, 1969*; *Geweke, 1982*; *Geweke, 1984*). Assuming that the two time-series are well described by Gaussian autoregressive processes, GC quantifies the ability to predict the future values of the time series of a given neuron using prior values of the time series of other neurons. In the field of neuroscience, GC has been used to analyze data from diverse sources (*Seth et al., 2015*), including electroencephalography (EEG), magnetoencephalography (MEG) (*Gow et al., 2008*), functional magnetic resonance imaging (fMRI) (*Roebroeck et al., 2005*; *Deshpande et al., 2009*; *Wen et al., 2013*), and local field potentials (LFP) (*Brovelli et al., 2004*). Despite criticisms such as sensitivity to data preprocessing (*Florin et al., 2010*), internal dynamics of the neuron (*Stokes and Purdon, 2017*), additive noise that can be correlated among regions of interest or not (*Nalatore et al., 2007*; *Vinck et al., 2015*), and sampling frequency (*Zhou et al., 2014*; *Barnett and Seth, 2017*), GC and its extensions (*Shojaie and Fox, 2021*; *Barnett et al., 2018*) have been useful to construct functional connectomes, to identify important neurons acting as information hubs, or to classify different brain states (*Nicolaou et al., 2012*).

While most GC analyses in neuroscience focus on large scale ensemble-level neuronal signals, it is unclear how well it can be adapted to address single-cell level descriptions, such as spiking (*Kim et al., 2011*; *Gerhard et al., 2013*; *Sheikhattar et al., 2018*) or population calcium imaging data (*De Vico Fallani et al., 2014*; *Severi et al., 2018*; *Oldfield et al., 2020*). Without the law of large numbers, these single-cell time series are no longer approximately Gaussian, the dynamics of signal propagation is increasingly nonlinear, and the sampling rate is typically much slower than the rate of information propagation. All of these points cast doubt on whether GC can correctly identify information flow. Recently however, *De Vico Fallani et al., 2014* demonstrated in zebrafish embryos that, despite a low sampling frequency of acquisition for calcium imaging (4 Hz), GC can detect information flow consistent with the known rostrocaudal propagation of activity among motoneurons in the spinal cord. *Oldfield et al., 2020* subsequently applied GC to grouped averages of calcium signals over large brain regions without single cell resolution, and detected the information flow correctly based on anatomy from the pretectum to the tectum in the visual system leading to subsequent activity of brainstem and motor circuits during prey capture in larval zebrafish. New variations of GC accounting for sparsity in the connection have also been developed and applied on single-cell level to both spiking and calcium signals in the primary auditory cortex of mice (*Sheikhattar et al., 2018*; *Francis et al., 2018*; *Francis et al., 2022*). Nonetheless, at the single-cell resolution, it is unclear to which extent the application of GC analysis in population calcium imaging is generalizable, and how GC analysis should be improved to accommodate intrinsic features of calcium signals.

In this manuscript, we identify a set of issues inherent to GC analysis when applied to population calcium imaging and propose solutions to address them. We first use synthetic neural signals and then exploit as examples population recordings of *in vivo* calcium signals in zebrafish with either correlated and non-correlated noise. We build an improved analysis pipeline customized for calcium signals to address critical issues intrinsic to *in vivo* calcium imaging, such as the slow exponential decays of calcium transients, correlated noise, and non-linear and non-Gaussian statistics. We develop our method both on a synthetic network of neurons for which we know the interaction structure, as well as on real data from motoneurons in the embryonic zebrafish spinal cord that are recruited along a chain organized from rostral to caudal on either side of the spinal cord (*De Vico Fallani et al., 2014*). We first improve our analysis pipeline to optimally detect the information flowing between embryonic motoneurons in *De Vico Fallani et al., 2014*. Second, we apply our improved GC analysis to infer information flow from large neuronal populations in the brainstem of larval zebrafish performing opto-motor response (*Severi et al., 2018*). In the context of large populations of neurons, we show that our method is able to reveal that motor-correlated neurons within the recently identified mesencephalic

locomotor region in **Carbo-Tano et al., 2022** drive the activity of downstream neurons in the reticular formation.

## Results

### Granger causality

Granger causality quantifies the ability to predict the future values of the discrete time series of neuron X of length $T$, $\{x_t\}$, for $t = 1, 2, \ldots, T$, using prior values of the time series of neuron Y, $\{y_t\}$, for $t = 1, 2, \ldots, T - 1$, assuming that the two time-series are well described by Gaussian autoregressive processes (**Granger, 1969**; **Seth et al., 2015**). The method asks whether including information from Y to predict X has significant predictive power, by computing a *p*-value assuming as a null model the probability that the dynamics of X can be explained without Y. The prior values of neuron X and Y itself used in that prediction are taken for a finite number of past time points, which defines a characteristic timescale.

With only the information of X itself, the dynamics of X, $x_t$, may be modeled as an autoregressive process, denoted as the '*reduced model*',

$$x_t = \widetilde{a}_0 + \sum_{q=1}^{L} \widetilde{a}_q x_{t-q} + \widetilde{\varepsilon}_t, \tag{1}$$

where $\widetilde{a}_q$ are the regression coefficients, the maximum lag $L$ is a hyper-parameter subject to tuning, and $\widetilde{\epsilon}_t$ is a residual term, assumed to be Gaussian.

Alternatively, one can also write down a linear regression model for $x_t$ given the history of both neuron X and neuron Y. This "*full model*" is

$$x_t = a_0 + \sum_{q=1}^{L} a_q x_{t-q} + \sum_{q=1}^{L} b_q y_{t-q} + \varepsilon_t, \tag{2}$$

where $a_q$ and $b_q$ are the regression coefficients for the contribution of neuron X and Y's history to neuron X's current state and $\epsilon_t$ is a residual term, also assumed to be Gaussian.

The key idea of Granger causality is to compare the residuals between the reduced and the full models. Intuitively, if neuron Y can influence X, including the dynamics of Y will improve the prediction of $x_t$, and hence reduce the residual term. As the reduced model is nested within the full model, to test whether the distributions of the residuals, $\{\varepsilon_t\}$ and $\{\widetilde{\varepsilon}_t\}$, are significantly different requires a one-way analysis of variance (ANOVA) test carried out by computing the *F*-test statistic:

$$F_{Y \to X} = \frac{\sum_{t=L+1}^{T} \left(\widetilde{\varepsilon}^2(t) - \varepsilon^2(t)\right)/(M_f - M_r)}{\sum_{t=L+1}^{T} \varepsilon^2(t)/(T_{regr} - M_f)}, \tag{3}$$

where $T_{regr} = T - L$ is the number of time points used in the regression, and $M_f = 2L + 1$ and $M_r = L + 1$ are respectively the number of parameters in the full and reduced models. Under the null hypothesis that including $y_t$ does not improve the prediction of $x_t$, and that the noise can be described by Gaussian i.i.d., the F-statistics will follow an F-distribution with defined degrees of freedom, $\mathcal{F}(F; M_f - M_r, T_{regr} - M_f)$. Hence, testing the computed F-statistic $F_{Y \to X}$ against the F-distribution will distinguish whether Y significantly Granger-causes X.

To measure the amount of reduction in the residual, the bivariate Granger causality value (BVGC) is defined as

$$\begin{aligned} GC_{Y \to X} &\equiv \max\left(\ln\frac{\text{var}(\widetilde{\varepsilon})}{\text{var}(\varepsilon)}, 0\right) \\ &= \max\left(\ln\frac{\sum_{t=L+1}^{T}\widetilde{\varepsilon}_t^2/(T_{regr} - M_r)}{\sum_{t=L+1}^{T}\varepsilon_t^2/(T_{regr} - M_f)}, 0\right), \end{aligned} \tag{4}$$

where the sample variance requires a correction using the number of samples in the regression ($T_{regr} = T - L$) minus the number of regression parameters ($M_r$ or $M_f$). Mathematically, the GC value is related to the *F*-test statistic through a monotonic transformation:

$$GC_{Y \to X} = \max\left(\ln\left[\left(\frac{M_f - M_r}{T_{regr} - M_f}F_{Y \to X} + 1\right)\frac{T_{regr} - M_f}{T_{regr} - M_r}\right], 0\right). \tag{5}$$

A significant-only GC value, $GC_{Y \to X}^{\text{sig}} = GC_{Y \to X}$ is called if the null test of the F-statistics is rejected, and zero otherwise.

In multi-component systems with more than two variables, one needs to consider the influence of other neurons or of a common stimulus, which we collectively denote by $\mathbf{Z} = \{Z_1, \ldots, Z_N\}$. For example, to analyze a time series of $N + 2$ neurons $\{x_t, y_t, \mathbf{z}_t\}$, one needs to distinguish direct information flow from Y to X, from indirect information flow from Y to Z and from Z to X, or from a common but delayed drive from Z to Y and from Z to X. **Geweke, 1984** extended Granger causality to a multivariate version (MVGC) to measure the conditional dependence between variables. In this case, we explicitly include the variables describing all possible other neurons and external stimuli, $\mathbf{z}_t = \{z_{1,t}, \ldots, z_{N,t}\}$, as regressed variables in both the reduced model and the full model. The current state of neuron X, $x_t$, now also depends on the past of Z, such that the reduced and the full models become

$$x_t = \widetilde{a}_0 + \sum_{q=1}^{L} \widetilde{a}_q x_{t-q} + \sum_{j=1}^{N} \sum_{q=1}^{L} \widetilde{c}_{j,q} z_{j,t-q} + \widetilde{\varepsilon}_t, \tag{6}$$

and

$$x_t = a_0 + \sum_{q=1}^{L} a_q x_{t-q} + \sum_{q=1}^{L} b_q y_{t-q} + \sum_{j=1}^{N} \sum_{q=1}^{L} c_{j,q} z_{j,t-q} + \varepsilon_t, \tag{7}$$

where $\widetilde{c}_{j,q}$ and $c_{j,q}$ are the regression coefficients from the conditioned signal $z_j$. The F-test statistics $F_{Y \to X|\mathbf{Z}}$ and the GC values $GC_{Y \to X|\mathbf{Z}}$, have identical expression as in **Equation 3** and **Equation 4**, with the number of parameters adjusted accordingly in the reduced and the full models. The significance test is adjusted using multiple comparisons by applying a Bonferroni correction to the $p$-value threshold.

It is important to point out that, although GC is based on linear regression of time series, its use is not limited to data generated by linear dynamics. In fact, as long as the data can be approximated by a vector autoregressive process, GC can be useful in revealing the information flow (**Seth et al., 2015**). For analysis of calcium signals, the intrinsic nonlinearity of calcium decay means it is not obvious a priori whether GC can be useful in revealing the information structure. As we will show in the following, the success of information retrieval depends on the details of the biological network, such as the interaction strength between pairs of neurons and the connection density, which leads to a complicated interplay among different timescales, such as the firing rate, the calcium decay time constant, and the time scale of information transfer.

## GC analysis of a small synthetic neuronal net

To gain some intuition about how GC identifies links, we test it on synthetic data generated using small networks of 10 neurons, where we pre-define the true connectivity (**Figure 1A, B**). One of the simplest types of dynamics is given by the same model that underlies the GC analysis, the vector autoregressive model (VAR), where the neural activities $f_{i,t}$ evolve according to

$$f_{i,t} = \sum_{q=1}^{L_{\text{true}}} \sum_{j} \Gamma_{ij,q} f_{j,t-q} + \xi_{i,t}, \tag{8}$$

where $\Gamma_{ij,q}$ is a memory kernel capturing the effect from neuron $j$ at time $t - q$ on neuron $i$ at time $t$, and $\xi_{i,t}$ a Gaussian noise with zero mean and variance $\sigma^2$. For simplicity, we assume the memory kernel acts only within the preceding two time steps, $L_{\text{true}} = 2$, and set $\Gamma_{ij,q} = A_{ij} s_j c$ for $q \leqslant L_{\text{true}}$, and zero, otherwise. $A_{ij}$ is the adjacency matrix: $A_{ij} = 1$ if neuron $j$ is directly pre-synaptic to neuron $i$, and $A_{ij} = 0$ otherwise. The sign of interaction $s_j$ shows whether neuron $j$ is excitatory ($s_j = 1$) or inhibitory ($s_j = -1$). The parameter $c$ is the connection strength, which we set to be positive. To ensure that the synthetic time series is stationary, we set the number of excitatory cells and inhibitory cells to be equal. The simulation time step is set as the unit time, $\Delta_{\text{sim}} = 1$.

We apply the GC analysis to synthetic data generated using the VAR model (**Figure 1C**) to see if we can recover the true network connectivity. For each neuron pair $(i, j)$, we compute the BVGC values following **Equation 4**, identifying $x_t = f_{i,t}$ and $y_t = f_{j,t}$, and computing the regression coefficients $a_q$ and $b_q$ using the standard least squares method. Analogously, we also compute the MVGC values, using all the remaining neuron signals $f_{k,t}|_{k \neq i,j}$ as the conditional variables $z_t$.

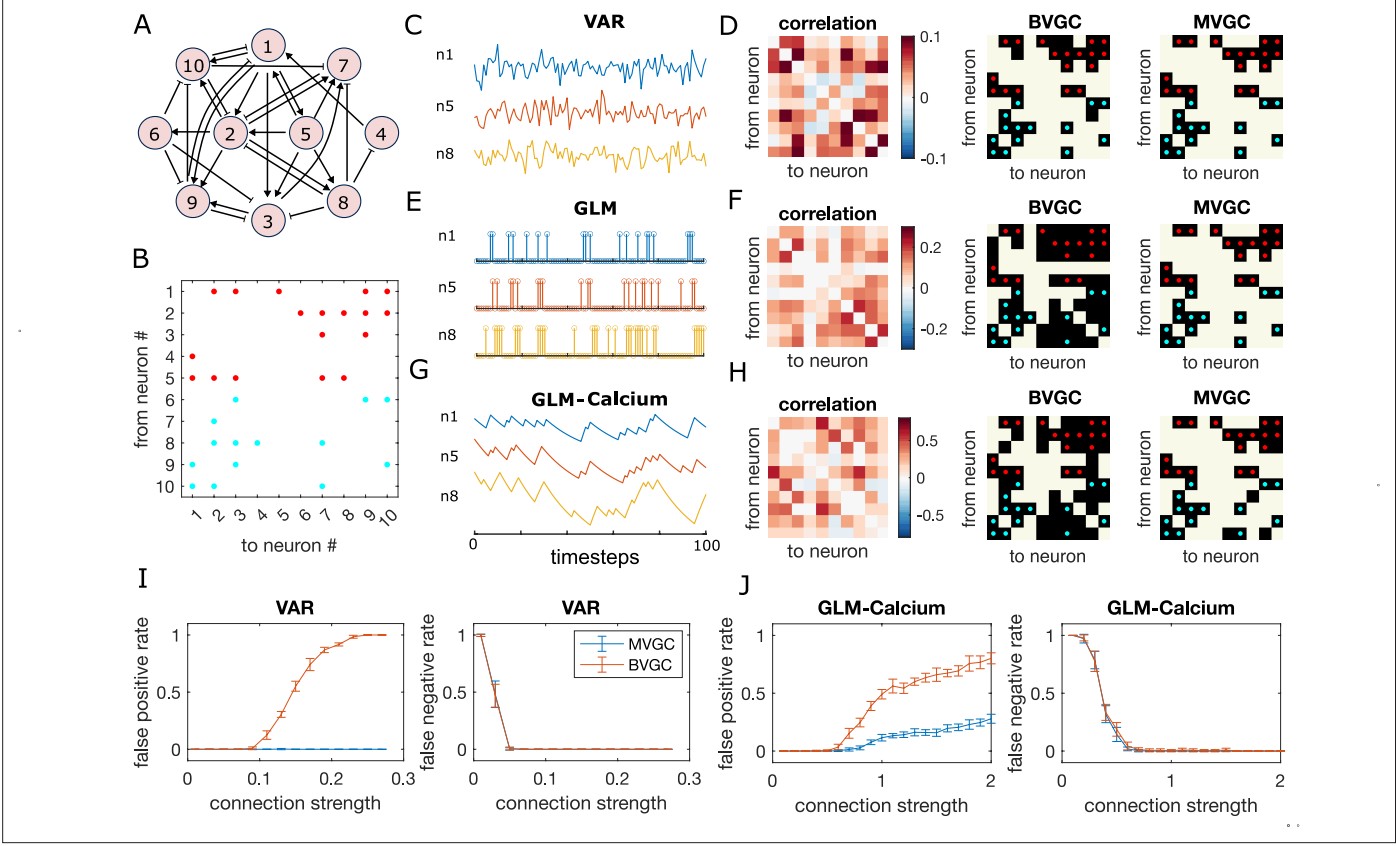

**Figure 1.** Granger causality (GC) analysis on synthetic data generated from *N* = 10 interconnected neurons of specific connectivity. (**A**) The network diagram and (**B**) the connectivity matrix of the model. The excitatory links are represented by arrows in (**A**) and red dots in (**B**). The inhibitory links are represented by short bars in (**A**) and blue dots in (**B**). Time series are generated and analyzed using correlation analysis and GC, using either a vector autoregressive (VAR) process with connection strength $c = 0.1265$ (**C–D**), a generalized linear model (GLM) with connection strength $c = 0.9$ (**E–F**), and a GLM with calcium filtering as in the experiments (**G–H**). (**D,F,H**) Comparison performance of correlation analysis using trajectory simulated with $T = 5000$ timepointes, bivariate (BVGC) and multivariate (MVGC) GC in predicting the connectivity matrix. True connectivity are indicated with red dots for excitatory links and blue dots for inhibitory links. The error rate of connectivity identification varies as a function of interaction strength $c$ for both the VAR dynamics (**I**) and the GLM-calcium-regressed dynamics (**J**). The multivariate GC always does better at predicting true connectivities than the bivariate (pairwise) GC and than correlation analysis. Error bars in (**I**) and (**J**) are standard deviation across 10 random realizations of the dynamics, each lasting $T = 5000$ timepoints.

The online version of this article includes the following figure supplement(s) for figure 1:

**Figure supplement 1.** The error rate of connectivity detection as a function of connection strength $c$ and the probability of connection $P_{\text{connect}}$, for VAR dynamics on randomly connected networks of 10 neurons.

**Figure supplement 2.** The error rate of connectivity identification as a function of trial duration and connection strength.

**Figure supplement 3.** The consistency of GC values as a function of trial duration and connection strength.

**Figure supplement 4.** Cross-correlation analysis for synthetic data generated with VAR dynamics reveals the correct interaction structure at the exact delay time difference.

**Figure supplement 5.** Cross-correlation analysis for synthetic data generated with GLM dynamics over-identifies connections.

**Figure supplement 6.** Cross-correlation analysis for synthetic data generated with the GLM-Calcium dynamics cannot identify the underlying structure of connectivity.

**Figure supplement 7.** Presence of redundant signals harms the performance of MVGC.

As shown in **Figure 1D**, MVGC successfully identifies all the links. However, the bivariate GC assigns spurious links, as it cannot distinguish direct links from indirect ones. In general, MVGC has a lower false positive rate (identified non-existing links) and false negative rate (missing true links) compared to BVGC when we vary the connection strength $c$ (**Figure 1I**). It is interesting to point out that, at small connection strengths, both MVGC and BVGC will fail to identify true links because the signal-to-noise ratio is too small to detect them. In comparison to both GC analyses, the equal-time correlation

completely misses the structure (see *Figure 1D* and Materials and Methods). While cross-correlation with the lag fixed to the model maximum lag $L_{\text{true}}$ performs better, it still fails to reproduce the true connectivity as faithfully as MVGC (see *Figure 1—figure supplement 4*).

To gain intuition if GC can work for more realistic single-neuron dynamics, which are typically very different from the linear VAR model, we perform the same analysis with dynamics generated with generalized linear models (GLM). We assume the spike count of neuron $i$ at time t, $\sigma_{i,t} \sim \text{Poiss}(\lambda_{i,t})$ is governed by a Poisson process with spiking rate $\lambda_{i,t}$.

$$\lambda_{i,t} = \exp\left(\mu_i + \sum_{q=1}^{L_{\text{true}}} \sum_{j} \Gamma_{ij,q} \sigma_{j,t-q}\right), \qquad (9)$$

where $\mu_i$ is the base rate for spiking, and the memory kernel $\Gamma_{ij,q}$ is identical to the VAR model. We can further convolve the spikes with an exponential decay function to simulate calcium signals (GLM-calcium; see Materials and Methods for details).

Although the GLM and GLM-calcium dynamics are highly non-linear and non-Gaussian (*Figure 1E and G*), MVGC successfully identifies true links (see *Figure 1FGJ*). In general, for large correlation strengths, the false negative rate (missing true links) is less than 0.1. Interestingly, for the GLM-calcium model, the false positive rate (identified non-existant links) at large connection strength is smaller than in the VAR model, as is the difference of errors between the MVGC and BVGC. A possible explanation is that for neurons with nonlinear interactions, the propagation of information in indirect links is less well described by linear models in direct links. In comparison, due to large auto-correlation caused by decay of the calcium signal, the correlation and cross-correlation performs much worse for the nonlinear dynamics than for the VAR model (see *Figure 1—figure supplement 5* and *Figure 1—figure supplement 6*).

This analysis of synthetic data shows that GC can reveal underlying connectivity structures from calcium signals, and that the performance difference between BVGC and MVGC is smaller in nonlinear models.

In the brain, a population of neurons can encode redundant signals. If the redundancy is in the structure of the network, for example, when two neurons share the identical input and output, the stochasticity will lead to different neuronal dynamics, and MVGC is able to identify the true connection. However, if two neurons have exactly the same activity, MVGC will underestimate the causality, while BVGC still reveal these connections (see *Figure 1—figure supplement 7*).

In GC analysis, it is essential to choose the maximum time lag $L$. If the maximum time lag $L$ is much less than the true interaction timescale $L_{\text{true}}$, then the reconstructed information flow will be incomplete. On the other hand, if $L$ is too large, we will run into the risk of overfitting. Here, we briefly discuss the effect of the maximum lag $L$ on the resulting GC network, using data generated by the synthetic network given by *Figure 1A*, simulated with VAR dynamics with true maximum lag $L_{\text{true}} = 3$.

As we vary $L$, the maximum time lag used in the GC regression models, we observe that increasing the maximum time lag does not always improve our ability to identify the network connectivity (*Figure 2A, B*). Going beyond the true maximum time lag leads to errors, increases both the false positive rate (*Figure 2*) and the false negative rate (*Figure 2B*). Compared to MVGC, BVGC is more prone to detect spurious links, even at the correct maximum time lag.

Interestingly, we observe that the average GC value as a function of the number of maximum lags can be used to identify the true maximum time lag $L_{\text{true}}$. As shown by *Figure 2*, both the average BVGC and MVGC values increase steeply as $L$ approaches the true maximum lag, and reach a plateau for a few maximum lag values before overfitting significantly increases the average GC values. Based on this observation, we select the maximum time lag to be the knee of the average GC value as a function of the maximum lag, as this maximum time lag captures enough information flow while the number of fitted parameters is kept as small as possible.

## GC analysis of motoneuron data
### The data
In order to test the performance of Granger causality on population calcium imaging data, we focus our analyses on previously published calcium transients in motoneurons of embryonic zebrafish (*Tg(s1020t:Gal4; UAS:GCaMP3)*, data published in *De Vico Fallani et al., 2014*), where the

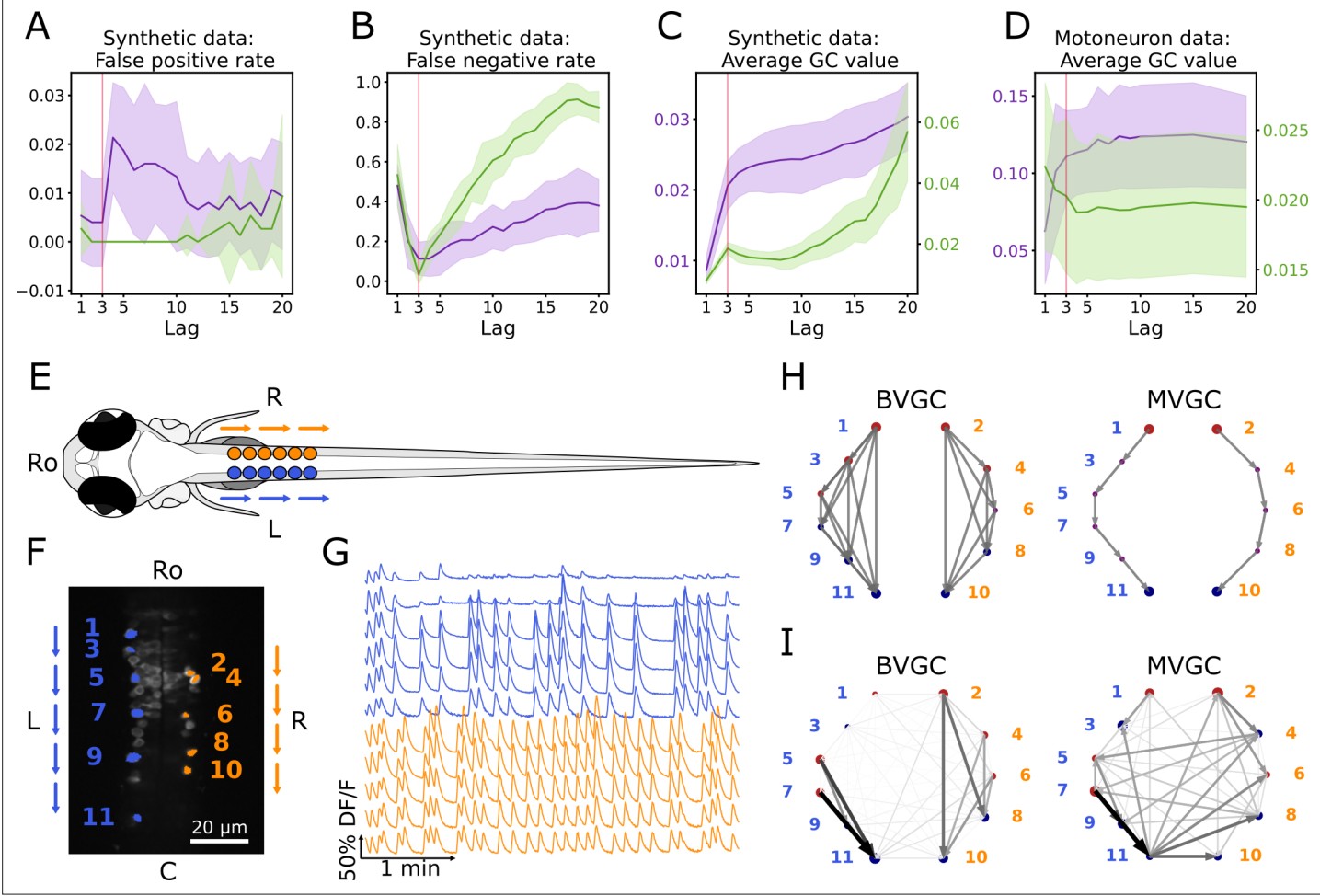

**Figure 2.** Bivariate and multivariate GC analyses of motoneuron calcium imaging recordings in a zebrafish embryo do not match expected ipsilateral and rostral to caudal information flow. (**A**) Effect of the maximum lag $L$ on the false positive rate, that is the proportion of spurious positive links over all negative links, on synthetic data (VAR) with a true lag of 3, for BVGC (*purple*) and MVGC *(green)*. BVGC is more prone to detect spurious links, even at the correct lag. (**B**) The false negative rate, that is the proportion of missed positive links over all positive links, is very sensitive to the choice of the lag. If the chosen lag is very different compared to the true lag, many links will not be detected. (**C,D**) Effect of the lag on the average GC values for synthetic (**C**) and real motoneuron (**D**) data ($N$ = 11 neurons, *De Vico Fallani et al., 2014*). Using the average GC value, we choose a lag of 3 for the motoneuron data analysis, corresponding to 750ms. Further supporting evidence for choosing $L = 3$ is given by the correlation analysis of GC values using different lags in *Figure 1*. Error bars in (A), (B) and (C) are standard deviation across 10 random realizations of the simulated dynamics with $T$ = 4000 on the network ($N$ = 10 neurons) specified by Figure1A. Error bars in (D) correspond to the standard deviation of the average GC value across the ten recordings of the motoneuron dataset. (**E**) Dorsal view of the embryonic zebrafish and positions of the left (**L**) and right (**R**) motoneurons targeted in calcium imaging (*colored circles*). The arrows represent the wave of activity, propagating ipsilaterally from the rostral (Ro) spinal cord to the caudal (C) spinal cord. (**F**) Mean image of a 30-somite stage embryonic zebrafish *Tg(s1020t:Gal4; UAS:GCaMP3)*, dataset *f3t2* in *De Vico Fallani et al., 2014*, with motoneurons selected for analysis shown in color. (**G**) Fluorescence signals (proxy for calcium activity) of selected neurons. We observe an oscillating pattern of activation of the left vs right motoneurons. (**H**) Representation of the motorneuron flow of information as a directed graph as would be ideally defined by bivariate (*left*) and multivariate (*right*) GC analyses. (**I**) Resulting directed networks for bivariate (*left*) and multivariate (*right*) GC analyses of fluorescence traces of (**G**) at maximum lag $L = 3$: we observe different networks than expected, both for BVGC and MVGC. This is consistent with the effect observed in (**A**), that BVGC detects many spurious links.

The online version of this article includes the following figure supplement(s) for figure 2:

**Figure supplement 1.** Comparison of Granger causality analysis results at different maximum lags.

**Figure supplement 2.** Validation of trial length sufficiency by high correlation of Granger causality values using the first and second halves of the motoneuron data.

underlying structure of information flow is known. The calcium transients were recorded using fluorescent tags *in vivo* for 250 s at 4 Hz and with single-cell resolution, in ventral spinal cord motoneurons while the zebrafish was showing spontaneous coiling activity of its tail (*Figure 2E–G*). During these fictive coiling events, the activity of motoneurons follows a wave from the rostral to the caudal side of the spinal cord (*Buss and Drapeau, 2001*; *Masino and Fetcho, 2005*; *Warp et al., 2012*). Upon activation along the rostrocaudal axis, spinal motoneurons that project to the axial skeletal muscle fibers elicit a rostrocaudal wave of muscle contractions that alternates between the left and right side of the body and propels fish forward. Accordingly, the calcium signals show steady oscillations alternating between the left and right sides of the spinal cord, leading to left-right muscle activation (*Figure 2G*). We use this data set in order to see if we recover the expected sequential information flow, from the rostral spinal cord to the caudal spinal cord (arrows in *Figure 2EF*, networks in *Figure 2H*).

To encode the underlying information structure of the neuronal network, we label the recorded $N$ neurons with index $i$, and the side of the neurons by $s_i \in \{\mathrm{L}, \mathrm{R}\}$. Assume that there are $N_l$ neurons in the left chain and $N_r$ neurons in the right chain, we order the indices such that the odd neurons are found on the left side, with $s_i = \mathrm{L}$ for $i = 1, 3, \ldots, 2N_l - 1$, and the neurons with $i = 2, 4, \ldots, 2N_r$ are found in the right chain with $s_i = \mathrm{R}$. The indices are then ordered by the neurons position along each of the two chains, such that if $s_i = s_j$, neuron $i$ is more rostral than neuron $j$ if $i < j$. The fluorescence signal for the $i$-th cell at time $t$ is denoted by $f_i(t)$, or equivalently by $DF_i/F_i(t)$ to represent the normalized fluorescence activity.

We will compute the GC network for multiple trials of calcium transients recorded in five fishes. Following the nomenclature of the dataset published in *De Vico Fallani et al., 2014*, we refer to the dataset recorded for fish $A$ and trial $B$ as *fAtB*; for example, *f3t2* denotes trial number 2 recorded for fish number 3. From the dataset, we exclude recordings for fish number 2 because the recorded time is 2 min, much less than that of the other recordings of 4 min.

## Naive GC

We apply the GC algorithm as described in Granger Causality to the calcium transients observed in embryonic motoneurons, and reproduce the results obtained in *De Vico Fallani et al., 2014* for the dataset *f3t2*. We choose the maximum time lag at the knee of average GC value, $L = 3$, corresponding to a duration of 750 ms, for both BVGC and MVGC analysis (*Figure 2D*). This knee corresponds to the maximum lag with the best balance between accuracy and complexity: further increasing the maximum lag yields very similar results but the GC analysis becomes much more computationally expensive (*Figure 2—figure supplement 1*). In order to verify that the length of the motoneuron dataset is sufficient to estimate the functional connectivity using Granger causality (see *Figure 1—figure supplement 2* and *Figure 1—figure supplement 3* for illustrations using synthetic data), we compared the two networks obtained by applying GC analysis to each half of the calcium imaging time series. The Pearson correlation coefficient between the GC values from the two halves is 0.87 for BVGC, and 0.75 for MVGC (see *Figure 2—figure supplement 2*), indicating the trial length is long enough for a consistent estimation of the GC values.

The GC network is then represented by directed graphs (*Figure 2I*). The nodes of the graph represent the motoneurons and the circle size is proportional to the *nodal delta centrality* as defined by *De Vico Fallani et al., 2014*: the larger the circle, the more central is the neuron in terms of its tendency to act as a transmitter hub (red color, positive value) or receiver hub (blue color, negative value) of information flow. Arrows display the functional connectivity links, that is the flow of information from one neuron to another. In the expected network assuming perfect information flow (*Figure 2H*), all the links are ipsilateral and represent rostrocaudal information flow.

Compared to *De Vico Fallani et al., 2014*, we identify the same functional connectivity matrices yielding the same connectivity networks. While the previous study focuses on recovering ipsilateral links and ignores contralateral links (depicted in dim gray in the network figures, see *De Vico Fallani et al., 2014*), we investigate all contralateral links and ipsilateral links (*Figure 2I*) as we aim to study why spurious contralateral connections occur. We represented the statistically significant direct GC links by arrows, proportional in size and color intensity to the GC value. The networks resulting from the GC analyses both for BVGC and MVGC show many other links than just rostrocaudal ones on either side.

In the MVGC analysis however, one would have expected only direct links to be significant, since conditioning on the other neuronal signals should prevent indirect links from being significant. This expectation is very different from the MVGC network in *Figure 2I*, where spurious contralateral links prevails.

Nonetheless, it is well known that MVGC is sensitive to additive noise, lack of data points, etc. To assess the impact of these artifacts on the resulting GC networks for motoneuron data, we define measures of information flows as following.

## Defining directional biases

The sequential and bi-axial information flow in the motoneuron network provides an excellent testing ground for GC. Specifically for the motoneuron datasets, to quantify the success of reconstructing the information flow in the motoneuron network, and to compare the information flow across different motoneuron datasets, we define the following measures of directional biases. First, we define the ratio of normalized total weights of ipsilateral links compared to all links

$$W_{\mathrm{IC}} = \frac{\langle G_{ij} \rangle_{(i,j)|s_i=s_j}}{\langle G_{ij} \rangle_{(i,j)|s_i=s_j} + \langle G_{ij} \rangle_{(i,j)|s_i \neq s_j}},$$ (10)

where $G_{ij}$ is the Granger causality values between neurons $i$ and $j$. If all connections are equal, then $W_{\mathrm{IC}} = 0.5$. We also define the ratio of ipsilateral rostral-to-caudal links compared to all ipsilateral links as

$$W_{\mathrm{RC}} = \frac{\langle G_{ij} \rangle_{i<j|s_i=s_j}}{\langle G_{ij} \rangle_{i<j|s_i=s_j} + \langle G_{ij} \rangle_{i>j|s_i=s_j}}.$$ (11)

This ratio gives a measure of the directionality of information flow. A bias in rostrocaudal links results in a departure from $W_{\mathrm{RC}} \sim 0.5$. Notice that based on the anatomy, the ground truth of information flow can be smaller than $W_{\mathrm{IC}} = 1$, because motoneuron activity has been shown to rely on command neurons in the brainstem that activate motoneurons sequentially in the spinal cord, not on motor neurons synapsing onto each other. With the two measures of directional biases of information flow, now we proceed to develop an improved pipeline for GC analysis. Using different datasets of the embryonic motoneurons as examples, we will show how to improve GC to analyze calcium transients of neuronal populations. Our goal is to recover a more biologically reasonable information flow compared to the naive GC network.

These measures are only constructed for the motoneuron networks. For more general networks with known structures, one can use the distance between the true and the GC-learned connectivity matrix to test the efficacy of the GC method. For networks with unknown structure, one can compare the GC-learned information flow by using different partitions of the data.

## Correcting for motion artifacts

In calcium imaging experiments, small movements of the animal body or the experimental setup can result in artificial fluorescence changes in the recording. In the calcium transients of data *f3t1*, we observe a big drop in fluorescence at a single time point out of 1000 (*Figure 3B* top panel) in the calcium traces of all neurons in the recording. As seen in the top panels *Figure 3C*, this is sufficient to corrupt the results of the GC analyses: neuron 10 (*Figure 3A*) appears to drive almost all other neurons, including contralateral neurons. We correct the motion artifact by setting the outlying time point's fluorescence value to the average of the previous and next values (*Figure 3B* bottom panel). The strongly dominant drive by neuron 10 disappears (*Figure 3C* bottom panels) and $W_{\mathrm{IC}}$ increases from 0.39 to 0.66 for BVGC and from 0.43 to 0.68 for MVGC (*Table 1*). This shows that GC analysis is sensitive to artifacts at single time points.

The motion artifact is a global noise on the observed neurons, a source that is well-known to create spurious GC links (*Vinck et al., 2015*; *Nalatore et al., 2007*). After careful examination of the calcium traces, we notice that the dominant neuron before motion-artifact correction, neuron 10, is characterized by its small dynamic range. Because the rest of the activity is low for neuron 10 and that a single time point seems to have a strong effect on other calcium traces, GC links driven by this neuron are high. We looked at other recordings containing a motion artifact and found that removing it did not

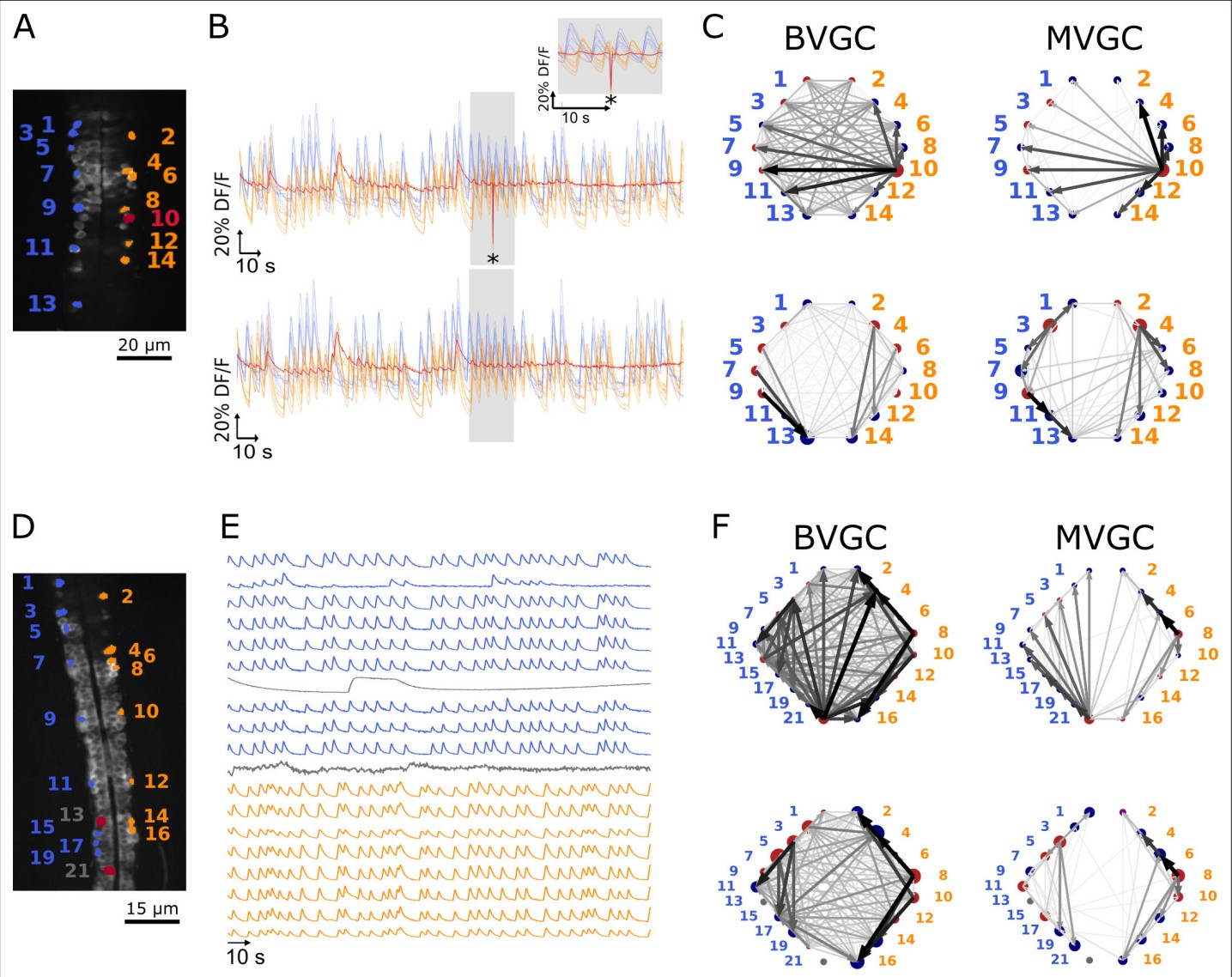

**Figure 3.** One time-point artifact or one noisy neuron fluorescence trace is sufficient to corrupt the GC algorithm. (**A**) Mean image of a 30-somite stage larval zebrafish *Tg(s1020t:Gal4; UAS:GCaMP3)*, dataset *f3t1* in ***De Vico Fallani et al., 2014***, with neurons selected for analysis shown in color. (**B**) Fluorescences traces of neurons of (**A**). *Top:* A motion artifact consisting of a drop in fluorescence at a single time point is present (star). *Bottom:* the motion artifact is corrected as being the mean of the two surrounding time points. (**C**) Directed graph resulting from the bivariate (*left*) and multivariate (*right*) GC analysis, before (*top*) and after (*bottom*) motion artifact correction. Before the correction, the neuron with the smallest activity appears to drive many other neurons, perhaps due to its lower SNR. This spurious dominant drive disappears once the motion artifact is corrected, demonstrating that GC is not resilient to artifacts as small as one time point out of one thousand. (**D**) Mean image of a 30-somite stage larval zebrafish *Tg(s1020t:Gal4; UAS:GCaMP3)*, dataset *f5t2* in ***De Vico Fallani et al., 2014***, with neurons selected for analysis shown in color. The fluorescence traces (**E**) of neurons displayed in red exhibit activity patterns different from the oscillatory pattern expected in motorneurons. GC analysis was run after removal of these neurons. (**F**) Directed graph resulting from the bivariate (*left*) and multivariate (*right*) GC analysis, before (*top*) and after (*bottom*) removing neuron 13 and neuron 21.

The online version of this article includes the following figure supplement(s) for figure 3:

**Figure supplement 1.** Re-scaled calcium transients exhibits a typical exponential decay with time constant $\tau_{ca} = 2.5$s, while atypical transients exhibit a slower decay.

always eliminate the spurious dominant neurons. The larger the artifact, the better we eliminate the spurious links: in ***Figure 3***, the artifact represents a deviation of about 80% of the *DF/F* and its impact on GC links is large. When the deviation is lower, the strong spurious links are not necessarily due to the motion artifact, therefore removing the artifact does not improve the expected network results.

**Table 1.** The directional measures for the bivariate and multivariate GC network for the calcium transients *f3t1* before and after motion-artifact correction, a pre-processing step for calcium signals. The GC network after motion artifact correction has clear ipsilateral structures.

| | BVGC | | MVGC | |
|---|---|---|---|---|
| | $W_{\text{IC}}$ | $W_{\text{RC}}$ | $W_{\text{IC}}$ | $W_{\text{RC}}$ |
| Before MA correction | 0.39 | 0.53 | 0.43 | 0.52 |
| After MA correction | 0.66 | 0.56 | 0.68 | 0.44 |

## Removing strange neurons

In order to improve the GC analysis results, we want to keep only clean calcium signals and remove the uncharacteristically behaving whose activities do not follow the stereotypical profiles of rise and falls. This is especially important for the MVGC analysis, in which traces of all other neurons are considered in the calculation of a GC value between two given neurons. Although BVGC values are not affected by neurons outside of the observed pair, having extra neurons might also influence the network structure through the GC values of the links.

In general, cells whose calcium transients showed time decay of tens of seconds were identified as non neuronal cells and excluded from GC analysis (see overlays of calcium transients in *Figure 3—figure supplement 1*). In the embryonic zebrafish traces *f5t2*, shown in *Figure 3E*, we remove neuron 13 (*Figure 3D*) that displays only one long calcium transient lasting for tens of seconds, and neuron 21, which does not exhibit an oscillating pattern.

GC networks are drawn for the neuronal populations before (*Figure 3F* top panels) and after removal of these outlier neurons (*Figure 3F* bottom panels). Before the removal, neuron 21 appears to drive many other neurons, likely due to its lower signal-to-noise ratio (SNR) and the correlated noise between calcium traces. Once removed, $W_{\text{RC}}$ increases, meaning that a mostly rostrocaudal propagation is observed, as information in motoneurons flows from the rostral to the caudal spinal cord (*Table 2*). We also observe a small decrease in $W_{\text{IC}}$ upon removal of neuron 21, due to the fact that the ipsilateral links going out from neuron 21 are not present anymore, but spurious contralateral links remain.

## Smoothing

One known limitation of traditional Granger causality analysis is its sensitivity to noise, especially correlated noise across variables (*Vinck et al., 2015*; *Nalatore et al., 2007*). In empirical calcium imaging data, correlated noise is common due to system-wise measurement noise and spurious correlations coming from the slow sampling rate.

To illustrate the problem of the noise, we return to the synthetic models in Granger Causality, a simulated system of 10-neurons with the VAR and the GLM-calcium dynamics, where the true underlying signal is $f_i(t)$, for each neuron $i$. We add a system-wide noise, such that the noise-corrupted signal is

$$g_i(t) = f_i(t) + \zeta(t), \tag{12}$$

**Table 2.** The directional measures for the bivariate and multivariate GC network for the calcium transients *f5t2* before and after removal of the atypical neurons.

| | BVGC | | MVGC | |
|---|---|---|---|---|
| | $W_{\text{IC}}$ | $W_{\text{RC}}$ | $W_{\text{IC}}$ | $W_{\text{RC}}$ |
| With strange neurons | 0.56 | 0.51 | 0.59 | 0.38 |
| Without strange neurons | 0.54 | 0.67 | 0.53 | 0.53 |

and $\zeta(t)$ is Gaussian white noise, with $\langle \zeta(t) \rangle = 0$, and $\langle \zeta(0)\zeta(t) \rangle = \sigma_\zeta^2 \delta(t)$. As shown in *Figure 4—figure supplement 2* and *Figure 4—figure supplement 3*, large correlated noise significantly decreases the ability of the Granger causality analysis to identify true connections.

We exploit the stereotypical shape of calcium signals – a fast onset and a slow decay – to parametrize the structure of the noise for empirical calcium signals. We assume that any correlation in the decay phase is the correlation of the noise. Additionally, we apply *total variation differentiation regularization* to denoise the fluorescence time series for each neuron (*Chen et al., 2019*) (see Materials and Methods and *Chartrand, 2011* for details). Briefly, for each neuronal signal $f(t)$, total variation differentiation regularization assumes the noise is Gaussian and white, and that the temporal variations in the derivative are exponentially distributed and only weekly correlated in time, which is well-suited to describe calcium data with episodes of continuous firing and continuous silence. Compared to usual total variation denoising that regulates the temporal variations in the signal, regulating the variation of the derivative is better at detecting sudden change of the signal strength, and is better at keeping the causality when applied to multivariate signals. As an example, *Figure 4* shows the original and the smoothed fluorescence signal for two neurons in the same fish (dataset *f3t2* from *De Vico Fallani et al., 2014*), as well as the residual noise. The noise in the motoneuron is correlated with correlation coefficients close to 1 for some pairs of neurons (*Figure 4B*). The large correlated noise is likely the cause of our problematic results in the MVGC analysis (see *Figure 2E* and *Figure 4C*). Denoising the calcium time series before the GC analysis eliminates links, especially in the MVGC approach (*Figure 4C*). The signal as quantified in terms of the relative weight of ipsilateral links $W_{IC}$ is also closer to the biological expectation ($W_{IC} \sim 1$) than before smoothing, with MVGC giving better results (*Figure 4D*).

We note that total variation regularization is a global denoising method that acts on the entire time series. Different from a low-pass filter, total variation regularization is able to keep the fine structure of the fast rise in the calcium signal, while smoothing out the noise. Nonetheless, the assumption is that the noise is Gaussian, which should be verified before smoothing the data by examining the high-frequency end of the signal's power spectrum. To address the problem of high-frequency correlated noise, one may propose to use spectral Granger causality (*Geweke, 1982*; *Geweke, 1984*), and only focus on the resulting GC values at low frequencies. However, we show that spectral GC applied to the unsmoothed calcium signal is unable to recover the information flow for any of the frequencies (*Figure 4-figure supplement 4*). Finally, in cases where correlated noise cannot be removed, there are alternative methods to reject false links, for example using Granger causality computed on the time-reversed data as the null hypothesis (*Vinck et al., 2015*).

## Slow calcium timescale

One potential pitfall of performing Granger causality analysis on single-cell calcium signals is that the timescale of the calcium indicator and the sampling are typically much slower than that of the information transfer. The onset of calcium signals can occur in few milliseconds, comparable with the timescale of information transfer, that is the time difference between the firing of a presynaptic neuron and a postsynaptic neuron downstream. Nonetheless, based on the intrinsic buffer capability of neurons and the fluorescent properties of the calcium indicator, calcium signals typically decay on the timescale of hundreds of milliseconds to seconds (*Tian et al., 2012*). In motoneurons, neurons can fire in bursts of spikes, which may further hinder the resolution of information propagation. In this section, we investigate how the interplay of these slow and fast timescales impacts the accuracy of Granger causality analysis to identify information flow.

In the embryonic spinal cord, motoneurons are recruited along the rostrocaudal axis sequentially. The muscle contractions alternate between the left and right side of the body, and propel fish forward. Correspondingly, the left and right sides of the motoneurons are active as a steady oscillation from the head to the tail, leading to an expected information flow from rostral to caudal (*Warp et al., 2012*). The spikes are followed by a plateau that lasts on the order of seconds (*Saint-Amant and Drapeau, 2001*). Using a combined statistical and dynamical approach, we simulate artificial "motoneuron" data (see example traces in *Figure 5A*), as two chains of five neurons, driven externally by two binary stimuli, $I_L(t)$ and $I_R(t)$, that randomly flip between an on ($I = 1$) and an off ($I = 0$) state, following the same statistics as recorded in the data. The stimulus of the left side, $I_L(t)$ is generated by randomly choosing time intervals when the stimulus state is on, $\tau_{on}$, and when it is off, $\tau_{off}$, uniformly from the

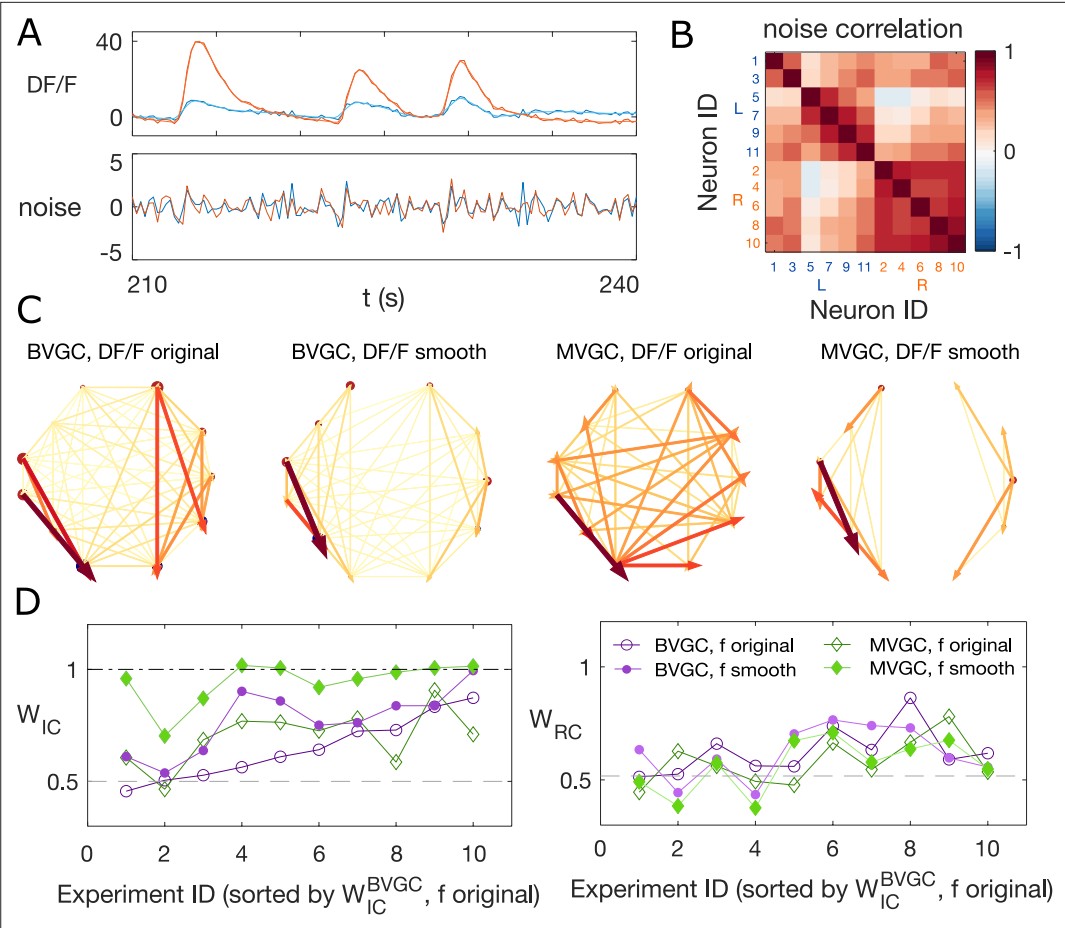

**Figure 4.** Smoothing the calcium imaging signal can improve the accuracy of GC. (**A**) We smooth the noisy calcium imaging signal $f = DF/F$ using total-variational regularization (see Materials and Methods), and plot the example traces of the original and the smoothened neuronal signals, and the residual noise, using the dataset *f3t2* ($N = 11$ neurons) from ***De Vico Fallani et al., 2014***. (**B**) The Pearson correlation coefficient shows the residual noise is correlated. (**C**) GC networks constructed using the original noisy calcium signal compared to ones using the smoothed signal for motorneurons. (**D**) Weight of ipsilateral GC links, $W_{IC}$, and the weight of rostal-to-cordal links, $W_{RC}$, for the original calcium signals and smoothened signals. We expect $W_{IC} \sim 1$ and a null model with no bias for where the links are placed will have $W_{IC} = 0.5$. The x-axis is sorted using the $W_{IC}$ value for bivariate GC analysis with the original calcium data. Data points are connected with lines to guide the eye. The bivariate GC results are plotted with purple circles, and the multivariate GC results with green diamonds. GC analysis results using the original noisy fluorescence signals are shown with empty markers, while the results from first smoothed data are shown with filled markers.

The online version of this article includes the following figure supplement(s) for figure 4:

**Figure supplement 1.** Cross-correlation of the residual noise after smoothing.

**Figure supplement 2.** Correlated noise increases the error of Granger causality analysis for synthetic data generated by VAR dynamics.

**Figure supplement 3.** Correlated noise increases the error of Granger causality analysis for synthetic data generated by GLM-calcium dynamics.

**Figure supplement 4.** Spectral Granger-causality does not improve information flow detection for the original, not pre-smoothened data.

---

sets $\left[\min(\tau_{on}^{data}), \max(\tau_{on}^{data})\right]$ and $\left[\min(\tau_{off}^{data}), \max(\tau_{off}^{data})\right]$, respectively. Here, $\{\tau_{on}^{data}\}$ is the set of duration of all "on"-states in the experiment (***De Vico Fallani et al., 2014***) that we find by the positive finite difference method in the fluorescent signal smoothed by total-variational regularization (***Chartrand,***

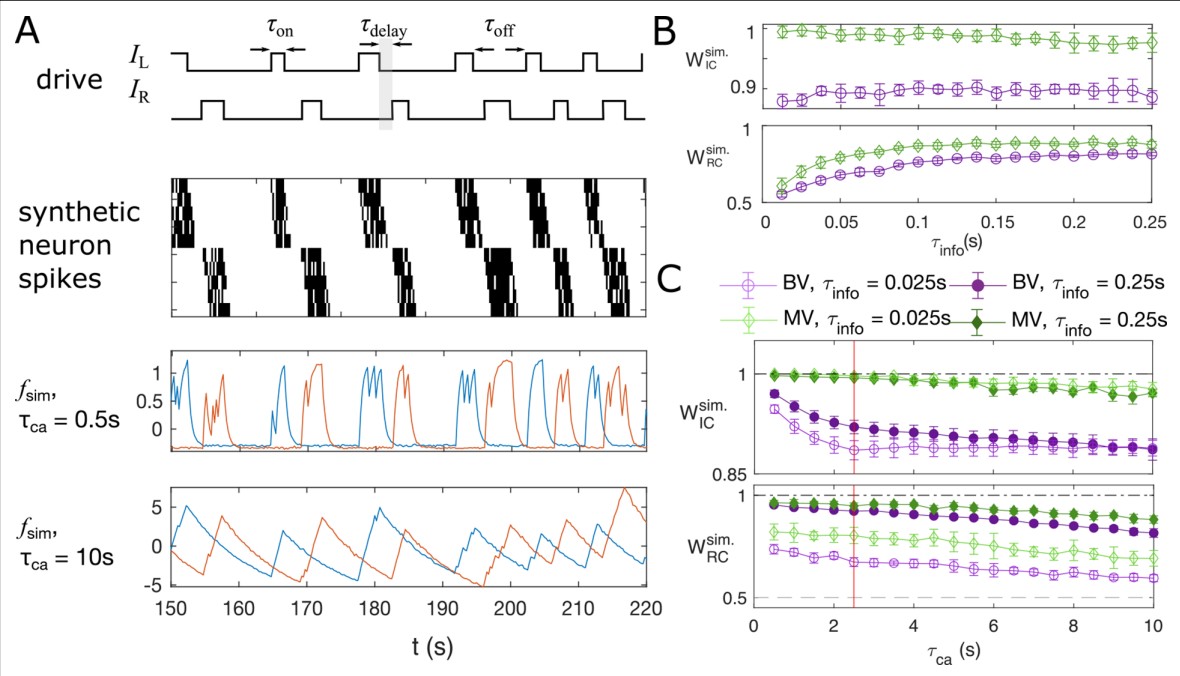

**Figure 5.** Synthetic data mimicking bursting motoneurons shows the slow timescale of calcium signal decay decreases GC performance. (**A**) Example traces of the synthetic data for two chains of five neurons ($N = 10$ neurons for the whole system). The common stimuli, $I_L$ and $I_R$, are sampled using the empirically observed on- and off-durations for the neuron bursts ($\tau_{on}$ and $\tau_{off}$). The time delay of the onset of the right stimulus $I_R$ from the offset of the left stimulus $I_L$ is also sampled using empirical distribution. Each synthetic data is generated for the duration of 1000s. (**B**) Success of information flow retrieval, measured by the weight of the ipsilateral GC links, $W_{IC}$, and the weight of rostral-to-caudal links, $W_{RC}$, as a function of the information propagation time scale $\tau_{info}$, evaluated at the empirical time scales $\tau_{ca} = 2.5s$, $\tau_{sampling} = 0.25s$, and biologically reasonable base spike rate $\lambda_0 = 32s^{-1}$. Results from bivariate GC are plotted in *purple*, and results from multivariate GC are in *green*. (**C**) Success of information flow retrieval for $\tau_{info} = 0.025s$ and $\tau_{info} = 0.25s$ for synthetic data with different calcium decay time constants. Greater calcium decay constants lead to worse information retrivals. Errorbars represents standard deviations across 10 realizations of synthetic data. The empirical calcium time scale $\tau_{ca} = 2.5s$ is indicated by the red vertical line. Error bars in panel (B) and (C) are standard deviation across 10 random realizations of the synthetic data.

2011; *Chen et al., 2019*). These sets of long-lasting time blocks, where the neurons are always active, simulates the plateau activity of the embryonic motoneurons. After $I_L(t)$ is sampled, the stimulus for the right-side neurons, $I_R(t)$, is generated such that the time delay between the "on"-state of $I_L(t)$ and $I_R(t)$ is sampled from $\left[\min(\tau_{delay}^{data}), \max(\tau_{delay}^{data})\right]$, where the delay time $\tau_{delay}$ matches the empirical stimulus distribution. We model the sequential activation of the motoneurons by imposing the spike rate as a function of $I_L(t)$ and $I_R(t)$ (see Materials and Methods) and we use a standard Poisson process with these rates to generate the spike train for each neuron, $\{\sigma_{i,t}\}$. The spikes are then convolved with an exponential decay with time scale $\tau_{ca}$ to simulate the slow decay time of calcium signals.

The timescales in this model include the sampling time $\tau_{sampling}$, the calcium decay time constant $\tau_{ca}$, and the timescale of information propagation $\tau_{info}$. The sampling time is given by the experimental setup for the motoneuron data we analyze (*De Vico Fallani et al., 2014*), with $\tau_{sampling} = 0.25s$. As shown in *Figure 3—figure supplement 1*, the empirical calcium decay time constant can be found by overlaying all epochs of decays of the calcium signals from different bursts and different neurons: fitting the decay of calcium signals to exponential functions gives $\tau_{ca} = 2.5s$. Notice this decay timescale is limited by both the buffering capability of the sensor, here GCaMP3, and the intrinsic buffering capability of the cell (*Tian et al., 2012*), that is having a faster sensor does not always decreases this decay time scale. The information propagation timescale is chosen such that $\tau_{info} < \tau_{sampling}$. The baseline spike rate is set to $\lambda_0 = 32s^{-1}$. Finally, the time resolution for the simulation is chosen to be $\Delta_{sim} = 0.0125s$, such that $\lambda\Delta_{sim} < 1$ to add stochasticity in the simulation.

Applying bivariate and multivariate GC analysis to the simulated data with the above parameters, we see that when the information propagation time scale is similar to the sampling time scale, both GC methods retrieve the correct information flow (*Figure 5B*). In general, multivariate GC performs

better in identifying information flow compared to bivariate GC. Even when the information propagation time scale is much smaller than that of the sampling rate ($\tau_{\text{info}} \leq \tau_{\text{sampling}}$), the information flow can still be correctly retrieved, and the multivariate GC gives the correct input $W_{\text{ipsi}}$ value 1, and $W_{\text{RC}}$ significantly larger than the null value of 0.5.

The success of retrieval of information flow also depends on the decay constant, although the effect is not strong. As shown in *Figure 5C*, the performance is almost perfect for short calcium decay times, with $W_{\text{IC}} = 1$ and $W_{\text{RC}} = 1$, and longer decay results in slightly worse performance. For the empirical calcium decay rate in the GCaMP3-expressing motoneuron data (indicated by the red line), $W_{\text{RC}}^{\text{MVGC}} = 0.81$. Naturally, using fast calcium indicators in the experiments facilitates the analysis, but the improvement may not be significant.

## Adaptive threshold for F-statistics

We now compute the Granger causality strength (*Equation 4*) on the smoothed calcium time series. Naively, the F-test is used to determine whether the Granger causality is significant (*Equation 3*). This test is exact if the residual, or the prediction error, is Gaussian-distributed and independent. For calcium signals, these assumptions do not hold: the noise is not necessarily Gaussian, and is not independent. While subsequent versions adapting GC to nonlinear and non-gaussian dynamics with assumptions of the underlying dynamics have been shown to have well-known asymptotic null distributions (*Kim et al., 2011*; *Sheikhattar et al., 2018*), it did not account for additional irregular stimuli. To develop a general method that can be applied to datasets with diverse stimuli, we use a data-driven method to construct the null distribution.

To test if we can still use the F-test as a test of significance, we generate data consistent with a null hypothesis by cyclically shuffling the time series of the driving neuron (see *Figure 6A*). Specifically, for the bivariate GC value from neuron $j$ to another neuron $i$, $GC_{j \rightarrow i}$, and the multivariate conditioned on the activity of the rest of neurons $\{f_k\}, k \neq i, j$, we compute the F-statistics with the time series $f_j(t)$ replaced by signal shuffled by a random time constant $\Delta t^{\text{rand}}$:

$$f_j^{\text{CS}}(t) = \begin{cases} f_j(t - \Delta t^{\text{rand}}), & 0 < t < T - \Delta t^{\text{rand}} \\ f_j(t + T - \Delta t^{\text{rand}}), & T - \Delta t^{\text{rand}} < t < T \end{cases}. \tag{13}$$

If the significance test remains valid for the calcium time series, we expect almost no significant links and the distribution of the F-statistics of the shuffled data should be the F-distribution. However, as *Figure 6BC* shows using the example dataset *f3t2* from *De Vico Fallani et al., 2014*, for both bivariate and multivariate GC, the distribution of the F-statistics for the shuffled data, $\mathcal{F}^{\text{shuffle}}$, has a shifted support that is larger than that of the F-distribution. This means that if we use the original F-test based on the F-distribution as a null model, even if there are no G-causal links between two neurons, we will classify the link as significant.

To improve specificity, we instead define the null expectation as the distribution for the F-statistics of the cyclically shuffled data to test for the significance of GC links. A GC link from the data is significant, if it has an F-statistics that is above the $1 - p$ percentile of the null F-statistics distribution, generated from the cyclically shuffled data. The null F-statistics distribution can be approximated non-parametrically by Monte Carlo reshuffles. However, since the significance threshold depends on the tail of the F-statistics distribution, to get an accurate estimation requires a large number of reshuffles and is computationally undesirable.

Instead, to accelerate the computation, we notice that the bulk of the $\mathcal{F}^{\text{shuffle}}$ distribution is well approximated by an F-distribution, with outliers explained by the pseudo-periodicity of the data. We parameterize the null empirical distribution by $\mathcal{F}(F; \alpha, \beta)$, and find the parameters by maximizing the likelihood (see Materials and methods for details). Repeating the significance test with the parametrized F-distribution, we find a resulting causality network with fewer links (*Figure 6D*).

## Pipeline summary and GC analysis of motoneuron data

Putting together all the observations of the previous sections, we developed an improved Granger causality analysis pipeline (*Figure 7A*) to overcome the limitations of naive GC for the investigation of directional information flow in neuron calcium data. These improvements consist of both

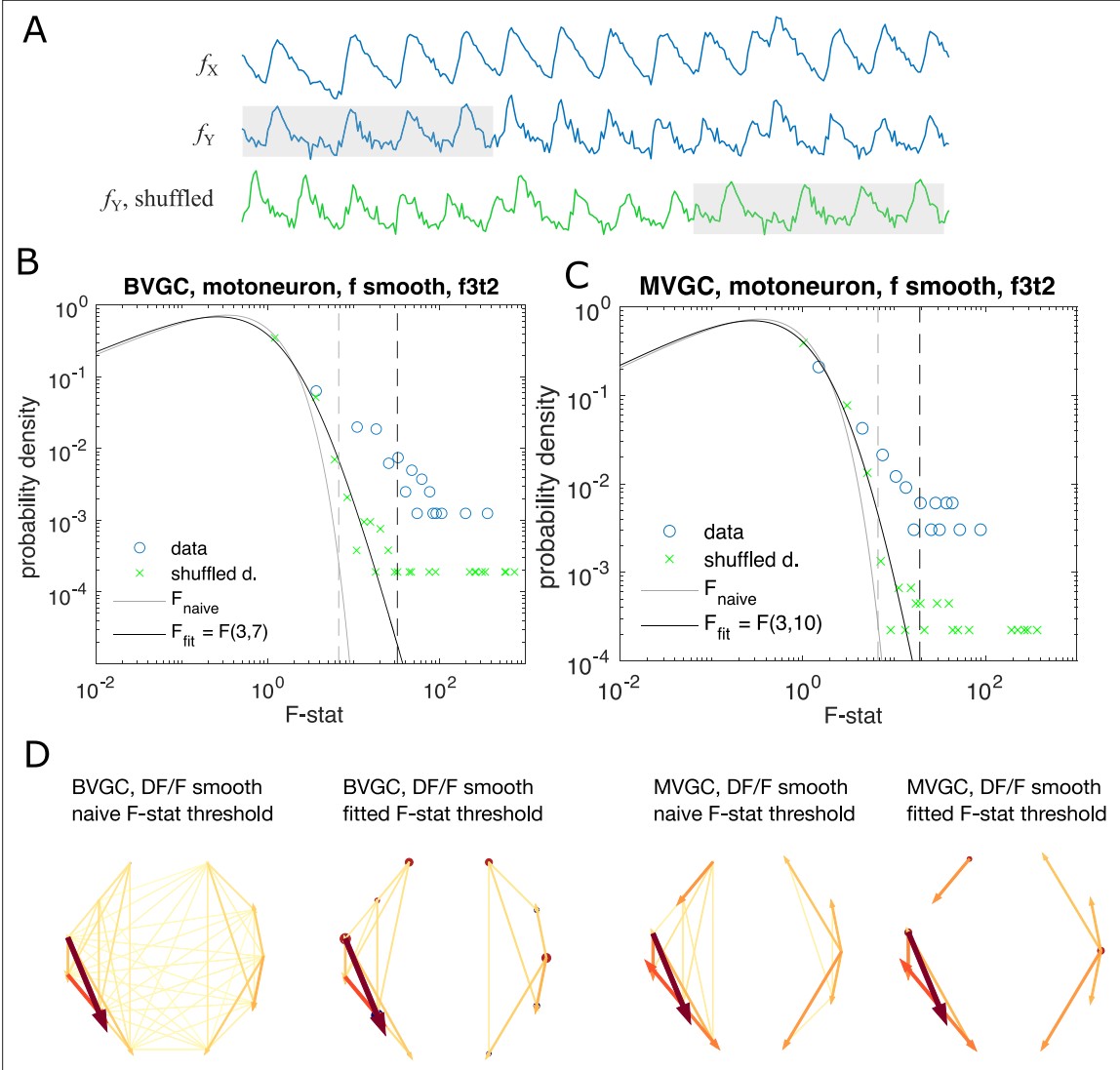

**Figure 6.** Significance tests for Granger causality on real calcium data require new thresholds, generated using null models. (**A**) Schematics for generating null models by randomly shuffling the data. The *blue* dots and curves correspond to the original data, the effect neuron $f_X$ and the cause neuron $f_Y$; the *green* ones are generated by random cyclic shuffling of the original $f_Y(t)$, with gray rectangles indicating matching time points before and after the cyclic shuffle. (**B**) The probability density of the *F*-statistics of the bivariate Granger causality for an example experiment (dataset *f3t2* from *De Vico Fallani et al., 2014* with *N* = 10 neurons). Blue circles are computed using the smoothed Calcium signals, and the green crosses are computed using the null model generated with cyclic shuffles. The black curve is the F-distribution used in the naive discrimination of GC statistics, and the gray curve is the best fitted F-distribution, fitting an effective number of samples. The gray and the black dashed vertical lines indicate the significance threshold for Granger causality, using the original threshold and the fitted threshold, respectively. (**C**) Same as (**B**), for multivariate Granger causality analysis. (**D**) Comparison between the significant GC network for the experiment *f3t2* using the naive vs. fitted F-statistics threshold.

pre-processing the calcium traces (see GC analysis of motoneuron data) and determining a custom threshold for the significance testing of the presence of a GC link, to avoid the many false positive links that the naive GC analysis finds (see Adaptive threshold for F-statistics). Pre-processing the data aims to remove undesirable effects of noisy fluorescence signals and of motion artifacts on the GC results. The first step consists of selecting only neurons displaying calcium traces that match their expectation (Removing strange neurons): calcium traces with a low signal-to-noise ratio or inconsistent behavior should be disregarded. This step is particularly important for MVGC analysis, in which the traces of all neurons are used to determine the presence of a link between any pair of neurons. Second, we correct the instantaneous change in fluorescence caused by motion artifacts (Correcting for motion artifacts). In the final pre-processing step, the calcium signals are smoothed to remove remaining

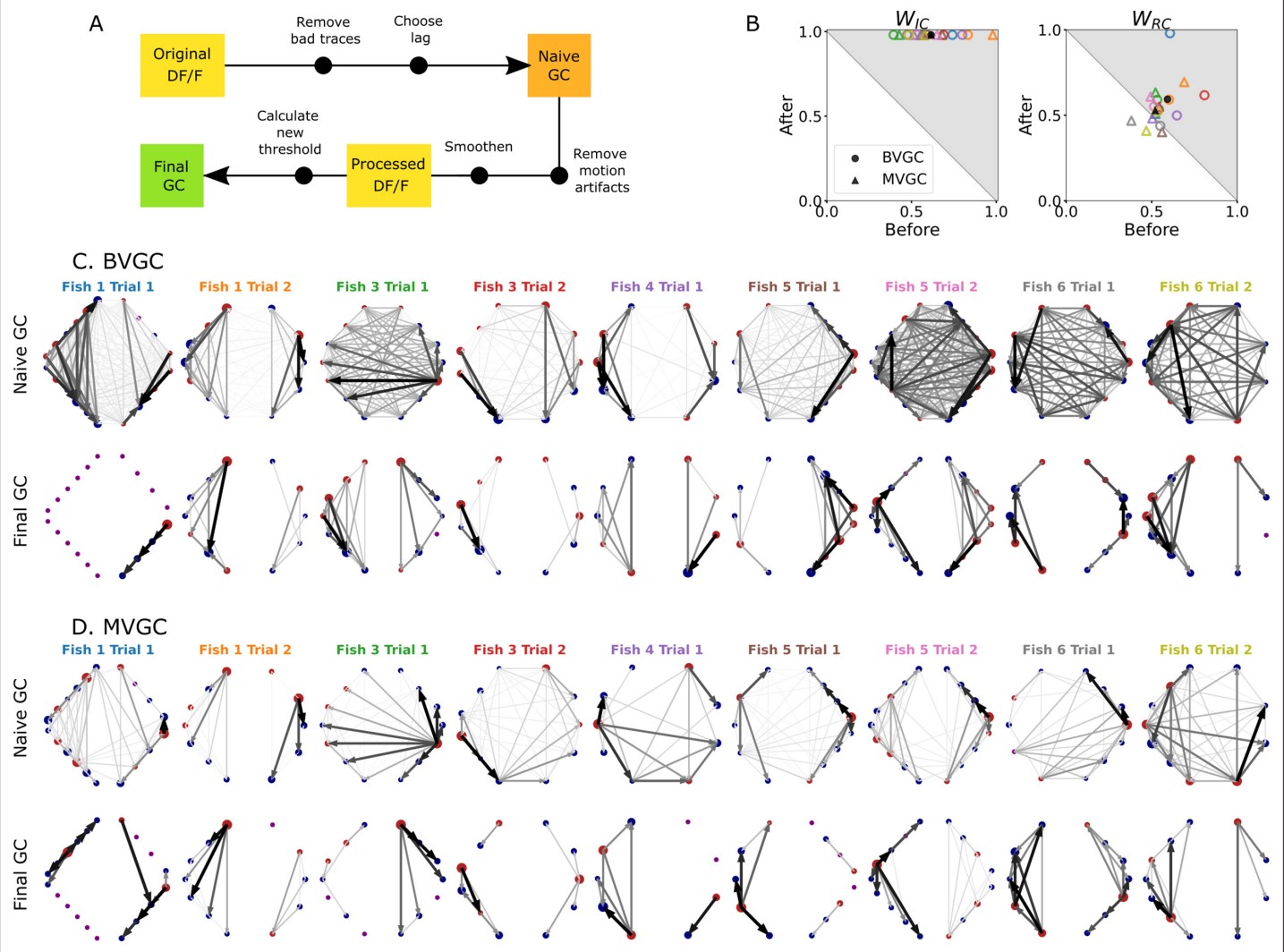

**Figure 7.** Comparison of GC analysis results with and without processing the calcium transients and adapting the threshold. (**A**) Description of the whole pipeline. Before running the GC analysis on the raw calcium transients ($DF/F$), the lag parameter must be chosen. Then, before running the GC analysis again, motion artifacts are corrected, the fluorescence is smoothed and a new threshold is calculated. (**B**) Comparison of $W_{IC}$ and $W_{RC}$ before and after applying the pipeline to the GC analyses. Points in the upper right triangle gray-shaded area represent the ratios that have increased in the final GC results. Each fish (n=9) is represented by a color, BVGC by circles and MVGC by triangles. Means are shown in black. We find that using our pipeline, $W_{IC}$ strongly increases and we can clear out the spurious contra-lateral links present in the original GC. $W_{RC}$ is not significantly improved: the recording frequency and GCaMP decay are likely too low and slow compared to the speed of rostro-caudal propagation of the information flow. (**C**) Network results before (*top row*) and after (*bottom row*) applying our pipeline for computing bivariate Granger causality (BVGC), for all fish. For consistency in the network representation and better ability to compare, we removed the uncharacteristically behaving neurons of the GC analysis before applying the pipeline. Note that more links are found on the side that was better in focus in the recording. (**D**) Same as (**C**) but for the multivariate Granger causality (MVGC).

noise (Smoothing), especially noise that is correlated across neurons and that leads to spurious links in the BVGC approach, and missing links in the MVGC analysis. In the BVGC analysis, correlation in the noise might be perceived as a link between two neurons. In the MVGC case, the correlation in the noise of the receiving neuron and in the noise of neurons on which the GC is conditioned can hide the actual influence of the signal of the driving neuron on the signal of the receiving neuron.

After these pre-processing steps, one can apply the GC algorithm to these calcium traces. In order to avoid calculating GC for many maximum time lags, the choice of this parameter can be chosen to match biological delay of information flow – here 750ms. For each pair of neurons, we time-shift one of the calcium traces in the GC analysis of a pair of neurons to remove causality between the two signals and use this new null model to determine the significance threshold, as motivated in

Adaptive threshold for F-statistics. Notice that for some datasets, the resulting MVGC network has more connections than the corresponding BVGC network. This is because the adaptive thresholds for the significance test are chosen separately for BVGC and MVGC analysis.

We take advantage of anatomical knowledge of the expected network for the motoneuron data set to compare the naive GC results to our customized GC results. The improved GC pipeline significantly increases the $W_{IC}$ ratio from $0.62 \pm 0.15$ to $1.00 \pm 0.00$ for both BVGC and MVGC ($p < 0.001$). $W_{IC} = 1$ indicates that there are no contralateral flow, corresponding to what is known from anatomy. On the other hand, the $W_{RC}$ ratio does not significantly increase ($0.59 \pm 0.09$ to $0.59 \pm 0.15$ for BVGC and $0.52 \pm 0.08$ to $0.53 \pm 0.10$ for MVGC, $p > 0.05$) (**Figure 7B**). $W_{RC}$ close to 0.5 shows that GC analysis is unable to capture the rostrocaudal information flow pattern.

The improvement of the ipsilateral ratio can be explained by the disappearance of most contralateral links that were spuriously detected by the naive GC analysis. The rostrocaudal ratio $W_{RC}$ is however not significantly improved. The temporal resolution of the recording (4 Hz leading to one frame every 250ms) may not be sufficient to correctly determine the directionality of the information flow as is much slower than the actual propagation speed (a few milliseconds, see **Masino and Fetcho, 2005**). In a similar manner, the decay time-constant of GCaMP3 is large compared to the refractory period between two spikes. Results could be further improved by recording at higher frequency and using novel fluorescent calcium sensors such as GCaMP6f, which has faster kinetics.

## GC analysis of hindbrain data

In order to investigate the information flow between neurons in a more complex dataset, we analyzed population recordings from neurons in the hindbrain of larval zebrafish performing optomotor response (OMR; data from **Severi et al., 2018**, also see **Figure 8A and B**), in which the true information flow is unknown. On each horizontal plane along the dorsoventral axis across the whole hindbrain, neurons were recorded *in vivo* using two-photon laser scanning calcium imaging at 5.81 Hz while *Tg(elavl3:G-CaMP5G)* transgenic larval zebrafish (previously known as *Tg(HUC:GCaMP5G)*) responded to a moving grading on their right side ($n = 139$ neurons in the reference plane, **Figure 8A**). Upon motion of the grading, zebrafish larvae responded by swimming forward, often with a left bias (**Figure 8C**, *right*). We classified hindbrain neurons in three categories: motor-correlated neurons in the medial V2a stripe (red in **Figure 8A and B**, **Kinkhabwala et al., 2011**; **Kimura et al., 2013**; **Pujala and Koyama, 2019**), other motor-correlated neurons (blue in **Figure 8A and B**) and non motor-correlated neurons (gray in **Figure 8A and B**). Because V2a neurons localized in the medial hindbrain are well-known command neurons driving locomotion (**Kimura et al., 2013**; **Pujala and Koyama, 2019**), we first focused our analyses on medial neurons whose activity was correlated to swimming (red neurons in the rectangle, $n_{med} = 20$, in **Figure 8A and B**). The corresponding calcium traces (**Figure 8B**) were exempt of motion artifacts and smoothing was not necessary as the scanning of the infrared laser leads to noise that is not correlated between neurons. When applying the naive bivariate Granger causality using the original threshold, we found almost all links to be significant (**Figure 8D**), suggesting numerous spurious links due to the high overall correlation in neuron activity. Motor-correlated neurons exhibited a higher drive (**Figure 8E**), defined as the sum of all its outgoing GC links:

$$\text{drive}_i = \sum_{j \neq i} GC_{i \to j} \tag{14}$$

where $j$ spans all other neurons. Critically, the neuronal drive was not correlated to the neuron's position along the scanning direction (correlation of $r = 0.15$ across all 139 neurons and $r = 0.03$ across medial swim-correlated neurons, $p > 0.05$, **Figure 8—figure supplement 1A**), implying that a correction for the scanning delay of the laser scanning microscope was not necessary. In contrast, the drive for a given neuron was correlated to the signal-to-noise ratio (SNR) of its calcium trace, computed following details in Materials and Methods, with a Pearson's correlation coefficient of $r = 0.45$ across all neurons and $r = 0.69$ across medial neurons ($p < 0.001$, **Figure 8E**). This effect could be corrected using an adaptive threshold tailored to each neuron pair to normalize the GC values. Interestingly, we noticed that the drive was higher for motor-correlated neurons, especially the ones located in the medial stripe and assumed to likely correspond to V2a reticulospinal neurons (**Kinkhabwala et al., 2011**; **Kimura et al., 2013**; **Pujala and Koyama, 2019**), further justifying focusing our attention on this subset of neurons.

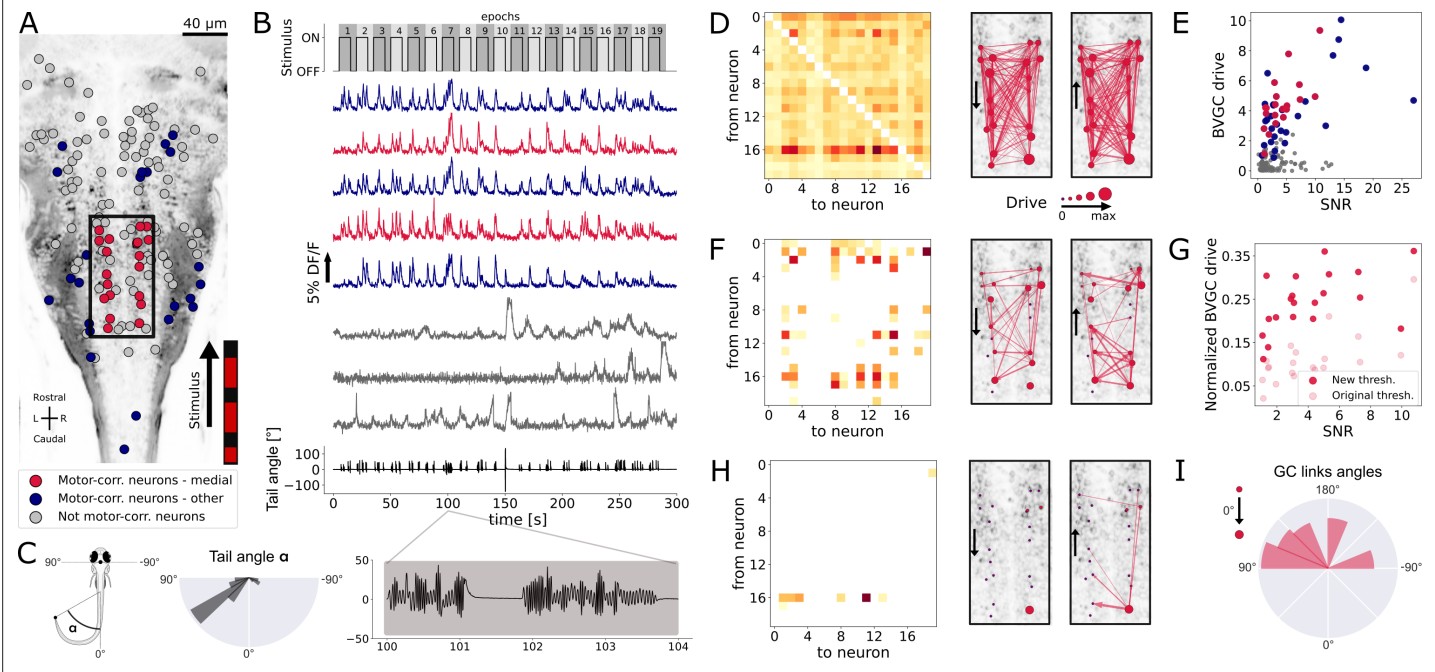

**Figure 8.** Our Granger causality pipeline revealed revealed right-to-left information flow among motor-correlated hindbrain neurons in larval zebrafish performing optomotor (OMR) responses. (**A**) Two-photon laser scanning calcium imaging *in vivo* was acquired at 5.81Hz in the hindbrain of *Tg(elavl3:GCaMP5G)* transgenic larvae performing OMR (data from *Severi et al., 2018*). Forty-one (*red and blue*) out of 139 neurons from a reference plane of a larval zebrafish hindbrain were motor-correlated and selected for subsequent analysis. Neurons located in the medial area (*red, N = 20 neurons*) are in the location of the V2a reticulospinal stripe (*Kinkhabwala et al., 2011*), distinct from other motor-correlated neurons (*blue, N = 21*). The activity of all other neurons (*gray, N = 98 neurons*) were not correlated to the motor output. (**B**) Sampled calcium traces of the three categories of neurons mentioned above were ordered by decreasing signal-to-noise ratio (SNR) from top to bottom. The moving grading (*top trace*) was presented to the right of the fish from tail to head to induce swimming, depicted as the tail angle (*bottom trace*, with a zoom over 4s showing two subsequent bouts, positive tail angles correspond to left bends). (**C**) The distribution of the tail angles for turns (tail angle $\alpha$ with $45° < |\alpha| < 90°$) was biased to the left. (**D**) GC matrix (*left*) and map of information flow (*right*, the arrows indicate the directionality of the GC links) for the naive BVGC on medial motor-correlated neurons. Of the 380 possible pairs, 351 were found to have a significant drive, suggesting that the naive BVGC algorithm is too permissive to false positive links, and justifying an adapted pipeline to remove spurious connections. (**E**) The neuronal drive, calculated as the sum of the strength of all outgoing GC links for each neuron, was correlated to the SNR of the calcium traces, especially high for medial neurons ($r = 0.69$, compared to $r = 0.45$ across all 139 neurons). (**F**) Customization of the threshold and normalization of the GC values reduced the number of significant GC links and revealed numerous commissural links that were both rostrocaudal and caudorostral between medial motor-correlated neurons. (**G**) The new neuronal drive was less correlated with the SNR (from 0.69 to 0.49 for medial neurons). (**H**) Due to the high correlation in the selected neurons and the relatively high number of neurons (20), the new MVGC pipeline highlighted only the strongest GC links. (**I**) The density distribution of the significant GC link angles from the new MVGC analyses revealed a biased right-to-left information flow. The directionality of the information flow is consistent with the presence of the stimulus on the right side of the fish and the fish swimming forward with a bias towards the left (C, *right*). These results were conserved across planes of the same fish as well as across fish (*Figure 4*).

The online version of this article includes the following figure supplement(s) for figure 8:

**Figure supplement 1.** Correlation of the drive and receiving values with the cell position and signal-to-noise ratio, and comparison across neuron groups.

**Figure supplement 2.** Distribution of the F-statistics of the shuffled data before and after re-scaling.

**Figure supplement 3.** Test for overfitting and information flow from regularized GC on zebrafish hindbrain.

**Figure supplement 4.** Distribution of GC link directions.

**Figure supplement 5.** Correlation of Granger causality values using the first and second halves of the hindbrain data.

To remove the spurious connections, we computed, as introduced before, the custom threshold for the significance testing of GC links. The difference from motoneuron dataset is that the activities of neurons in the hindbrain are driven by a periodic stimulus. Therefore, to generate the null statistics, we randomize the neural activities of the driver neurons by shuffling the neural signals, while keeping the stimulus fixed. More specifically, we divide the entire recording into 19 epochs of consecutive onset (10 s) and offset (5 s) of the stimulus (see *Figure 8B*), and generate shuffled calcium traces by

randomizing the order of these 19 epochs. For each neuron pair $(i, j)$, we apply $m$ such random shuffles to the calcium trace of the driver neuron $i$, and compute the F-statistics $\{F_{i \to j}^{\text{shuffled}}\} = \{F_{i \to j}^{s_1}, \ldots, F_{i \to j}^{s_m}\}$. For BVGC, $m = 1000$, and for MVGC, $m = 100$, we denote the empirical distribution of $\{F_{i \to j}^{\text{shuffled}}\}$ by $\mathcal{F}^{\text{shuffled}}$.

As in the previous examples, we need to parameterize $\mathcal{F}_{i \to j}^{\text{shuffled}}$. As before, for each neuron pair $(i, j)$, we noticed that the distribution $\mathcal{F}_{i \to j}^{\text{shuffled}}$ was well-described by a constant-rescaled F-distribution (*Figure 8—figure supplement 2*). Applying an adaptive threshold on $F_{i \to j}$ reduces to applying the original threshold on the normalized $\widetilde{F}_{i \to j}$, defined by dividing the naive $F$-statistics with the expectation value of the $F$-statistics generated by the shuffled data. Mathematically,

$$\widetilde{F}_{i \to j} = F_{i \to j} / \langle F_{i \to j}^{\text{shuffled}} \rangle. \tag{15}$$

Finally, by analogy, we use *Equation 5* to compute the normalized version of the GC values, directly from the normalized F-statistics. The final BVGC matrix is shown in *Figure 8F*. Normalizing the BVGC values allowed the reduction of the correlation between the drive and the SNR, from 0.69 to now 0.49 for medial motor-correlated neurons (*Figure 8G*). With this new threshold for significance, most spurious links disappeared. The remaining significant BVGC links between medial motor-correlated neurons revealed strong information flow between these key command neurons. The MVGC resulted in a much smaller number of significant links (*Figure 8H*), suggesting a strong global correlation between the neurons, with only the strongest links remaining.

In order to quantify the direction of global information flow, we computed the distribution of the angles of the significant GC links. We normalized the raw distribution by the distribution of all possible GC links in the network, to suppress the influence of neuron positions on the shape of the histogram (see Materials and methods). The normalized distribution (*Figure 8I*) revealed a biased right-to-left information flow. Because the visual grading was presented on the right side of the fish, moving forward from tail to head, one expects the fish to turn more to the left. As expected, we observe in the plane of interest a biased distribution of the tail angles towards the left (*Figure 8C*, *right*). Such right-to-left flow of information was systematic across all motor-correlated neurons in the same plane (*Figure 8*, *blue*), as well as across the 11 planes in the same reference fish, and for all planes of the 10 fish (*Figure 8—figure supplement 4*, *blue, purple, gray respectively*).

The observed flow of information from right to left in the hindbrain could reflect the recruitment of excitatory command neurons in order to recruit spinal motor neurons on the left, and start swimming with a left bend. Upstream of command neurons lies a critical high motor center that has been identified by electrical stimulations across multiple vertebrate species and is referred to as the mesencephalic locomotor region, or MLR (*Shik et al., 1966*). We recently identified, functionally and anatomically, the locus of the MLR in larval zebrafish (*Carbo-Tano et al., 2022*). We therefore investigated the GC links across three ventral and horizontal planes where numerous command neurons are located in the caudal hindbrain (*Kimura et al., 2013*; *Pujala and Koyama, 2019*): the reference plane analyzed in *Figure 8*, and the adjacent planes respectively 10 µm more dorsal and more ventral than the reference plane (*Figure 9A–C*, *left*). In the behavioral recordings acquired for all three planes, we observed a biased distribution of the tail angles towards the left (*Figure 9A–C*, *top right*). Accordingly, a right-to-left flow of information was found across motor-correlated neurons (*Figure 9A–C*, *bottom right*). Remarkably, we also detected strong driver neurons (*Figure 9A–C*, *left*) in the locus of the right MLR centered in rhombomere 1 (*Carbo-Tano et al., 2022*, *Figure 9B*, *left*). As expected from anatomy (*Carbo-Tano et al., 2022*) and physiology (*Brocard et al., 2010*, *Cabelguen et al., 2003*, *Ryczko and Dubuc, 2013*, *Ryczko et al., 2016*), neurons in the right MLR locus drove the activity of neurons in the left MLR locus as well as more caudally located neurons in the ipsilateral and contralateral hindbrain.

## Discussion

Granger causality is a simple and effective method for information flow retrieval in large brain areas (*Shojaie and Fox, 2021*; *Barnett et al., 2018*; *Nicolaou et al., 2012*). However, to apply GC analysis on calcium signals from population of neurons at single-cell level requires a careful re-examination of the method, because of the non-linearity, non-Gaussianity, and the possible predominance of correlated noise in the data depending on the imaging method used. We have developed an improved analysis pipeline and demonstrated its usefulness in retrieving information flow in zebrafish

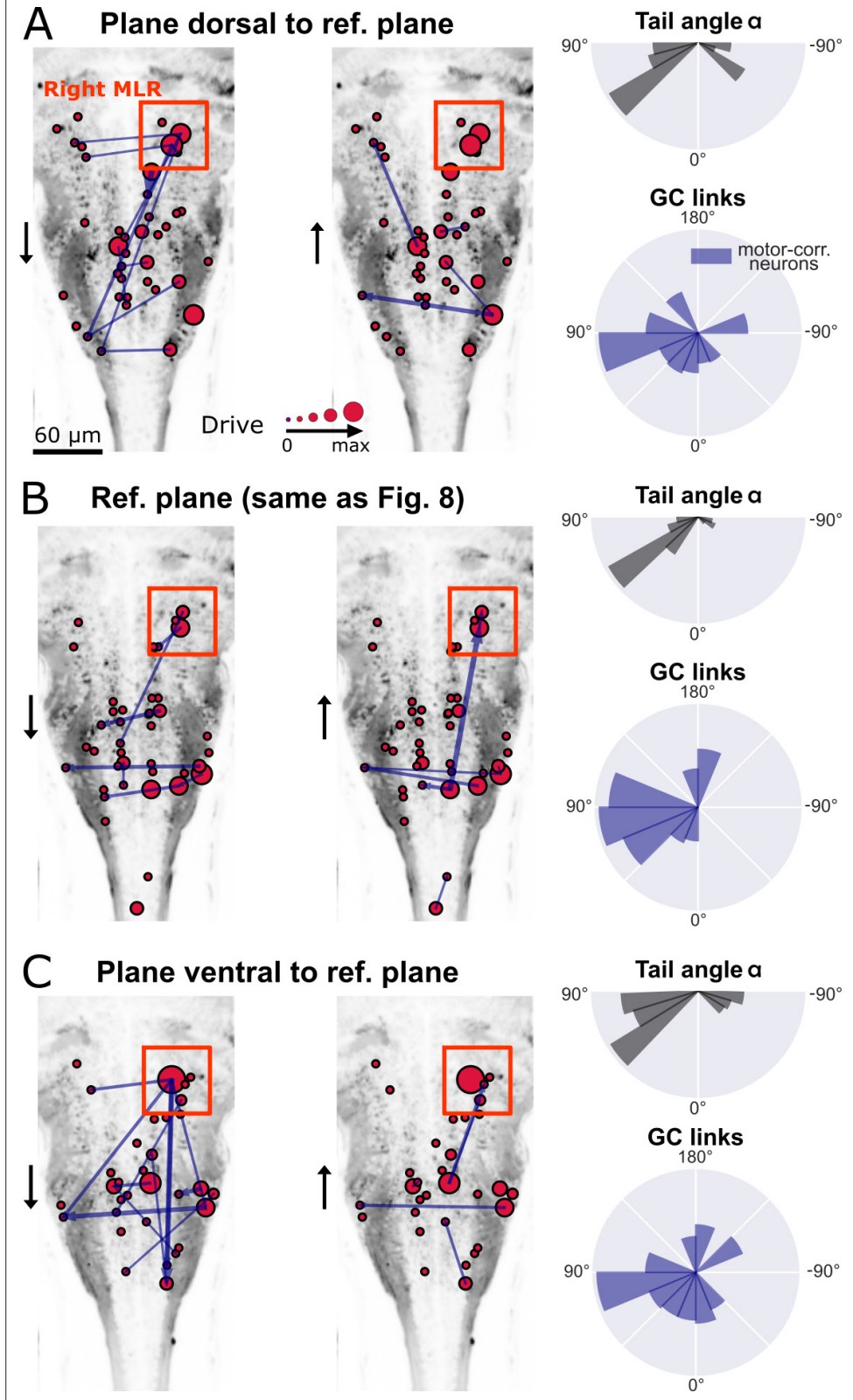

**Figure 9.** New multivariate Granger causality analysis of motor-correlated hindbrain neurons revealed neurons in the locus of Mesencephalic Locomotor Region (MLR) on the side of the visual grading driving activity of neurons in the contralateral MLR and on both sides in the medulla. The distributions of the tail angles for turns (tail angle $\alpha$ with $45° < |\alpha| < 90°$, *gray, top left* of **A-C**) and the distributions of the new MVGC link angles (*blue, bottom*

*Figure 9 continued on next page*

*Figure 9 continued*

*left* of **A-C**) indicate a bias when the visual grading stimulus was presented on the right towards the left across 3 hindbrain planes spaced by 10μm (**A**: plane 5, dorsal to **B**: plane 7, analyzed in *Figure 8*; **C**: plane 9, ventral to plane 7). Across these three planes, a group of strong driver neurons (*middle and right panels* of **A-C**) was found located on the right, in the locus of the MLR. This is to our knowledge the first evidence of recruitment of cells in the MLR locus during locomotion. As expected from anatomy (*Carbo-Tano et al., 2022*), right MLR neurons drive contralateral neurons in the left MLR and in the medulla where the reticular formation is located, both on the right and left sides.

---

either within a dozen of motoneurons (*De Vico Fallani et al., 2014*) and tens of hindbrain neurons (*Oldfield et al., 2020*). Since the properties of calcium signals in neuronal populations are comparable across animal models, the tailored analysis pipeline we are proposing is not limited to zebrafish, and can be applied to calcium imaging data across animal species.

The first concern of applying GC analysis to calcium signals is how well GC, a measure of linear dependence among signals, can retrieve information flow in these non-linear and non-Gaussian imaging data. Using synthetic data on small networks with different types of dynamics mimicking calcium signals, from linear autoregressive models to spiking dynamics with exponential decay, we have demonstrated that GC can successfully reconstruct the underlying true dynamical interactions among neurons for a wide range of dynamical variables. In the case of motoneurons that fire repeatedly in bursts, we modeled network activity with two chains of neurons using empirically identified dynamical parameters. In this simulation, we showed that GC can retrieve the expected ipsilateral flow of information with rostral-to-caudal directionality, even when the information propagation occurs at a faster scale than the experimental sampling time and the time decay of the calcium transient. Altogether our results on simulations show that GC is a useful analysis method for population recordings of single-cell level calcium signals.

Throughout the examples used, we compared results from bivariate GC and multivariate GC analysis. While BVGC overestimates the true number of causal links, we find that MVGC is prone to a winner-take-all phenomenon that may represent just one of many plausible system-level models that can account for the observed data. This is especially problematic when the signals are strongly correlated, for example, in the presence of redundant signals. In addition, MVGC is more prone to the issue of overfitting. We therefore suggest using BVGC on datasets with large numbers (hundreds) of neurons, and MVGC on datasets with few tens of neurons. In general, starting with a small number of neurons of special interest seems wise.

After improving the confidence on the GC method, we applied it to real calcium signals. Because real data is noisy and noise can be correlated or not depending on the imaging method used, GC analysis requires careful examination of multiple pitfalls originating from these calcium signals that we addressed in our analysis pipeline. Our pipeline illustrates the pre-processing of the signals and the post-processing of the GC results. This effort is complementary to previous studies where new variations of GC were introduced to take into account the non-linear and non-gaussian dynamics in interacting neurons, for example by modifying the underlying VAR dynamics with GLM or VAR with sparsity constraints, and subsequently using adaptive thresholds for significance tests (*Kim et al., 2011*; *Sheikhattar et al., 2018*; *Francis et al., 2018*; *Francis et al., 2022*). Future work of interest should combine these variations of GC with the pipeline we built here.

1. *Preprocessing of calcium imaging data*. As reported previously in *Vinck et al., 2015*; *Nalatore et al., 2007*, we find that GC analysis is sensitive to correlated noise, which can often occur in calcium signals, due to, for example, motion artifacts or fluctuations of the imaging laser for non-scanning imaging. We highlight that a motion artifact occurring at a single time point can create spurious GC links and should therefore be removed before proceeding to GC analysis. Similarly, correlated noise across neurons throughout the measurement will also results in spurious links, and will be especially problematic for multivariate GC analysis. The pitfalls associated with correlated noise can be resolved by identifying statistical features of the noise during the decay of calcium signals, and subsequently removing them using total variation differentiation regularization as we used here, or other nonlinear filtering methods. Future versions of the smoothing should be applied to all neurons simultaneously, assuming that the noise is correlated, to best preserve the time-ordering of the signal.

2. *Adaptive threshold for significance test in GC analysis*. The naive GC analysis uses a significance test that compares the F-statistics with the null test, a F-distribution. While this null model is exact for Gaussian processes, for real data with nonlinear dynamics, non-Gaussian noise or data with periodic stimuli, the null distribution often deviates and needs to be learned through randomization of the data. In order to develop a more stringent test of significance for GC links, we generated null models of the F-statistics using random cyclic shuffles of the real data. By using a pair-specific threshold for the F-statistics, we were able to correct for the effect of the signal-to-noise ratio on the GC value.

When real calcium traces were acquired with a spinning disk microscope, that is using a camera chip for detection of all pixels at once illuminated by a laser, the noise was correlated because all pixels over the field of view were recorded simultaneously. In this case, smoothing the fluorescence traces was consequently necessary. In contrast, this step is not advised with two-photon laser scanning microscopy or high speed volumetric light-sheet imaging in which respectively different pixels and planes are simultaneously scanned across the sample. Furthermore, we found that the scanning delay in laser scanning microscopy acquired at 5.81 Hz did not influence the GC results: neurons recorded at the beginning of the scanning cycle did not appear as driver neurons of neurons recorded at the end of the cycle. We found however that the drive of a given neuron computed from the naive GC was correlated with the signal-to-noise ratio (SNR) of the calcium trace. Using an adaptive threshold for the significance test, the correlation between the drive and SNR decreased.

Applied to motoneuron data, the new pipeline led to an improvement in removing the spurious contralateral links, but did not significantly improve the directionality of the information flow from rostral to caudal — likely due to the small temporal resolution of the data (4 Hz) while activity propagates in few tens of milliseconds between segments.

In population recordings of brainstem neurons with unknown underlying connectivity, we first focused on neurons whose activity was correlated with the motor output and were located in the area of important command neurons, referred to as the V2a medial stripe (*Kinkhabwala et al., 2011*) that was previously shown to be critical for generating motor output (*Kimura et al., 2013*). We found that V2a neurons whose activity was correlated to the motor output were more likely to drive the activity of other neurons than neurons whose activity did not encode the motor output. In the condition where the stimulus is presented on one side of the fish, resulting in the fish often performing contralateral turns, the adapted multivariate Granger causality analysis revealed that the information flows mostly from the ipsilateral to the contralateral side in the brainstem. In order to further understand the origin of the drive underlying this effect, we extended the analysis to all neurons that had correlated activity with the motor output, including neurons outside the V2a medial stripe. Remarkably, we found a cluster of neurons that contained strong driver neurons on the side where the visual stimulus was presented and in the locus of the MLR, a motor center that recruits neurons in the V2a medial stripe to generate motor output (*Carbo-Tano et al., 2022*). Our observations suggest that MLR cells on the ipsilateral side mainly drive the activity of contralateral MLR cells and motor-correlated neurons to elicit swimming. Throughout vertebrate species, the MLR has been defined functionally based on electrical stimulations effective at eliciting movement, but had not been yet identified from neuronal recordings during locomotion due to the difficulty to access this deep region in the brainstem during active locomotion. To our knowledge, our result is therefore the first evidence in vertebrates of recruitment of cells in the MLR locus during active locomotion.

We have focused in this manuscript on issues that are specific to calcium imaging. We only briefly discuss problems that are also common to other neural data types such as fMRI. Firstly, we emphasize that with current experimental techniques, only a small portion of the entire nervous system is recorded. While newer versions of GC have been adapted to take into consideration latent variables which contribute equally to all neurons (*Guo et al., 2008*), or through subsequent removal of causal links (*Verny et al., 2017*), GC does not in general allow identification of the true underlying connection in the presence of unobserved neurons. However, improved GC is able to identify information flow and an effective functional causal network for the observed neurons. Another issue specific to multivariate GC is the problem of overfitting (*Seth et al., 2015*), especially when the number of neurons is large compare to the number of independent samples in the data. In this situation, one would need to consider group the activities of neurons (*Meshulam et al., 2019*), either according to their correlation, or by adjacency, and apply GC to this coarse-grained network. One can also apply GC to subnetworks of observed neurons, and construct a distribution of information flows. Note that other approaches to

identify causal links than GC have been developed and applied in genomic data (**Verny et al., 2017**), gene regulatory networks (**Meinshausen et al., 2016**) and large-scale neural networks (**Ke et al., 2019**). Further work is now needed to explore whether and how these approaches could be adapted with our improved pipeline to calcium recordings.

Overall, comparing to common practices of using correlation to extract a functional network from calcium neural signals, our GC pipeline provides more information by informing on the direction of information flow. Additionally, compared to existing methods to overcome the pitfalls of calcium data, such as addressing the global additive noise using an inverse-time GC (**Vinck et al., 2015**), our pipeline preprocesses calcium signals taking advantage of their slow decay, and keeps the GC approach as a simple method. By writing down GC value as a function of the F-statistics, normalized by null models directly constructed using shuffled data, we were able to correct the GC network by adjusting the values of individual links, and thereby to reduce the influence of signal-to-noise ratio in a simple yet effective way. The improved pipeline makes it possible to apply Granger causality analysis to data from experiments that simultaneously record from high numbers of neurons in larger networks.

## Materials and methods

### Synthetic dynamics on small networks

We simulate artificial neural networks with three different dynamics, evolving at a time resolution of $\Delta_{\text{sim}}$. For simplicity, we set the information flow to occur at the same time delay. The memory kernel, or the influence of neuron $j$ on neuron $i$ separated by time lag of $q$ (in the unit of $\Delta_{\text{sim}}$), is denoted as $\Gamma_{ij,q}$. The three dynamics are:

1. *Vector autoregression models* (VAR)

$$f_{i,t} = \sum_{q=1}^{L_{\text{true}}} \sum_j \Gamma_{ij,q} f_{j,t-q} + \xi_{i,t},$$

   where $\xi_{i,t}$ is a white noise. The vector autoregression model is identical to the underlying model for Granger causality analysis, and allows GC to best reproduce the network structure.

2. *Spike dynamics for generalized linear model* (GLM)
   We simulate spiking neurons with non-linear interaction is through the generalized linear model (GLM), where the spiking rates of the neurons are given by

$$\lambda_{i,t} = \exp\left(\mu_i + \sum_{q=1}^{L_{\text{true}}} \sum_j \Gamma_{ij,q} \sigma_{j,t-q}\right)$$

   and the spike counts

$$\sigma_{i,t} \sim \text{Poiss}(\lambda_{i,t}).$$

   Here, $\mu_i$ gives a base rate for spiking, $\Gamma_{ij,q}$ is how the spike rates depend on the neurons' past history. The spike counts $\sigma_{i,t}$ are what we consider as synthetic data, and what we will use for Granger causality analysis.

3. *Spike dynamics regressed by exponential decay* (GLM-Calcium)

   To simulate Calcium neuronal data, which has a fast onset and a slow decay, we add an exponential regression to the spike simulation with the GLM. The resulting trace is

$$f_{i,t} = \sum_{q=0}^{\infty} \sigma_{i,t-q} e^{-q/\tau_{\text{ca}}} \Delta_{\text{sim}}.$$

We simulate the above three dynamics with the interaction network, $\Gamma_{ij,q}$. The network has equal numbers of excitatory and inhibitory neurons to ensure that the dynamics is stationary. For convenience, the strength of the connection is set to be identical, $\Gamma_{ij,q}|_{q<L_{\text{true}}} = cA_{ij}$, where $c$ is the connection strength, and $A_{ij}$ is the signed adjacency matrix.

### Cross-correlation

The simplest approach of finding links in networks is through cross-correlation,

$$C_{ij}(\Delta t) = \frac{\sum_t (f_i(t) - \bar{f}_i)(f_j(t + \Delta t) - \bar{f}_j)}{\sigma_{f_i} \sigma_{f_j}}.$$

(16)

The (equal-time) correlation matrix given by $C_{ij}(\Delta t = 0)$ is commonly used in calcium signal analysis as a functional network. Nonetheless, it is symmetric, and does not reveal directional information.

## Smooth calcium trace using total variation regularization

We follow the same procedure as in *Chen et al., 2019* to smooth calcium traces using total variation regularization. Existing MATLAB packages and custom code are used to reconstruct the smooth trace *Chartrand, 2011*. Here, we outline this smoothing method briefly.

Consider a raw calcium fluorescence signal $f(t)$ from neurons that are known to generate episodic spikes and silences. These episodic activities can be characterized by the underlying derivative signal $u(t)$ which governs the onset and offset of the episodes, which we assume have an exponentially distributed temporal variations, and is only weakly correlated in time. If we further assume that the noise in $f$ is Gaussian and white, the maximum likelihood reconstruction of the signal $f(t)$ is equivalent to minimizing.

$$F(u) = \frac{\tau_f}{\sigma_f} \int_0^T dt |\dot{u}| + \frac{1}{2\sigma_n^2 \tau_n} \int_0^T dt |Au - f|^2 \,, \tag{17}$$

where $A$ is the antiderivative operator, the combination $\sigma_n^2 \tau_n$ is the spectral density of noise floor that we can retrieve from the power spectrum of $f$ at high frequencies, $\sigma_f$ is the total standard deviation of the signal, and $\tau_f$ is the typical time scale of these variations. We determine the one unknown parameter $\tau_f$ by asking that, after smoothing, the cumulative power spectrum of the residue $Au - f$ has the least root mean square difference from the cumulative power spectrum of the extrapolated white noise.

## Compute signal-to-noise ratio

We compute the signal-to-noise ratio (SNR) for the fluorescence trace of a neuron, $f(t)$, using the smoothing procedure above. Namely, for each trace $f(t)$, we first apply the smoothing procedure. Then, the power of the noise, $P_{\text{noise}}$, is given by the extrapolated white noise, and equivalently by the sum of the power spectrum of the residue $Au - f$. The sum of the power spectrum of $f$ gives the sum of the power of the noise and the power of the signal, $P_{\text{signal}} + P_{\text{noise}}$. Finally, the SNR is computed as the ratio of $P_{\text{signal}}/P_{\text{noise}}$.

## Synthetic statistical data generation

To study the effect of slow calcium decays on the accuracy of GC analysis, we generate synthetic data statistically. Specifically, we consider two chains of five neurons, driven externally by block stimuli with progressing time-delay across each chain, and alternates between the left and the right side. The stimulus of the left-side is generated by choosing intervals, $\tau_{\text{on}}$, for consecutive on-state (and $\tau_{\text{off}}$ for consecutive off-states) uniformly from the set $\left[\min(\tau_{\text{on}}^{\text{data}}), \max(\tau_{\text{on}}^{\text{data}})\right]$, where the $\{\tau_{\text{on}}^{\text{data}}\}$ is the set of duration of all 'on'-states in a real zebrafish experiment, founded by positive finite difference in the fluorescent signal smoothened by total-variational regularization. The stimulus of the right-side follows with the delay time $\tau_{\text{delay}}$ also matches the uniform distribution on the support of the empirical distribution.

We model the sequential activation of the motoneurons by imposing the spike rate as

$$\lambda_{L_k}(t) = I_L(t - k\tau_{\text{info}})\lambda_0,$$
$$\lambda_{R_k}(t) = I_R(t - k\tau_{\text{info}})\lambda_0,$$

where $k$ is the ordered index of neurons on each side, $\tau_{\text{info}}$ is the time scale of information propagation, and $\lambda_0$ is a large basal spike rate to simulate the plateau in the neural activity. We use a standard Poisson process with these rates to generate the spike train for each neuron $\{\sigma_i, t\}$. Finally, the observed calcium signal is computed by convoluting the spikes with exponential decays for each neuron $i$,

$$f_{i,t} = \sum_{q=0}^{\infty} \sigma_{i,t-q} e^{-q/\tau_{\text{ca}}} \Delta_{\text{sim}}$$

and downsampling at a sampling frequency $\tau_{\text{sampling}}^{-1}$.

## Neuron identification in the hindbrain dataset

We analyzed the raw fluorescence movies from *Severi et al., 2018* using the *suite2p* software (*Pachitariu et al., 2017*), a pipeline for processing two-photon calcium imaging data. The pipeline first performs 2D registration of the plane recording to remove motion artifacts. Regions of interest (ROIs) are detected automatically, classified into *cell* and *non-cell*, and their calcium traces are extracted by the software. Following the automatic neuron classification, we performed a manual verification to remove spurious cells and add missing cells. These were selected based on the shape of the detected ROI and on the shape of the extracted fluorescence trace. The V2a neurons are identified by their positions and motor-correlated calcium activitites. The MLR neurons are identified based on their location within the MLR locus as determined in *Carbo-Tano et al., 2022*, and specifically not using any information about their shape nor the fact that their activity is motor-correlated.

## Normalized polar histogram of Granger causality links

We computed the distribution of the angles of the significant GC links to quantify the direction of global information flow. To suppress the influence of neuron positions on the direction distribution, we normalized the distribution by dividing it over the distribution of the angles of all possible connections given the position of the neurons.

## Acknowledgements

We thank Urs Böhm and Kristen Severi for providing feedback on the hindbrain data, Jenna Sternberg for the data on motoneurons, and Fabrizio De Vico Fallani, Mario Chavez (Paris Brain Institute) and Moritz Grosse-Wentrup (University of Wien) for insightful feedback. This work was supported by the European Research Council Consolidator Grants n. 724208 (AMW) and n. 101002870 (CW), the New York Stem Cell Foundation (NYSCF) Robertson Award 2016 #NYSCF-R-NI39 (CW), the Human Frontier Science Program (HFSP) Research Grant #RGP0063/2018 (CW), the Fondation pour la Recherche Médicale Team grant (FRM- EQU202003010612) and the Fondation Bettencourt-Schueller #FBS-don-0031 (CW).

## Additional information

### Competing interests

Aleksandra M Walczak: Senior editor, eLife. Claire Wyart: Reviewing editor, eLife. The other authors declare that no competing interests exist.

### Funding

| Funder | Grant reference number | Author |
|---|---|---|
| European Research Council | COG 724208 | Aleksandra M Walczak |
| European Research Council | COG 101002870 | Claire Wyart |
| New York Stem Cell Foundation | NYSCF-R-NI39 | Claire Wyart |
| Human Frontier Science Program | RGP0063/2018 | Claire Wyart |
| Fondation pour la Recherche Médicale | FRM- EQU202003010612 | Claire Wyart |
| Fondation Bettencourt-Schueller | FBS-don-0031 | Claire Wyart |

The funders had no role in study design, data collection and interpretation, or the decision to submit the work for publication.

## Author contributions

Xiaowen Chen, Conceptualization, Data curation, Software, Formal analysis, Investigation, Visualization, Methodology, Writing – original draft, Writing – review and editing; Faustine Ginoux, Conceptualization, Data curation, Software, Investigation, Visualization, Methodology, Writing – original draft, Writing – review and editing; Martin Carbo-Tano, Validation, Investigation; Thierry Mora, Conceptualization, Supervision, Investigation, Methodology, Writing – original draft, Project administration, Writing – review and editing; Aleksandra M Walczak, Conceptualization, Supervision, Funding acquisition, Investigation, Methodology, Writing – original draft, Project administration, Writing – review and editing; Claire Wyart, Conceptualization, Resources, Data curation, Supervision, Funding acquisition, Investigation, Visualization, Methodology, Writing – original draft, Writing – review and editing

## Author ORCIDs

Xiaowen Chen ⓘ http://orcid.org/0000-0002-4029-1805
Faustine Ginoux ⓘ http://orcid.org/0000-0001-5031-7346
Martin Carbo-Tano ⓘ http://orcid.org/0000-0002-1936-7174
Thierry Mora ⓘ http://orcid.org/0000-0002-5456-9361
Aleksandra M Walczak ⓘ http://orcid.org/0000-0002-2686-5702
Claire Wyart ⓘ http://orcid.org/0000-0002-1668-4975

## Decision letter and Author response

Decision letter https://doi.org/10.7554/eLife.81279.sa1
Author response https://doi.org/10.7554/eLife.81279.sa2

# Additional files

## Supplementary files

• MDAR checklist

## Data availability

The current manuscript is a computational study, so no data have been generated for this manuscript. All data used in our manuscript have been previously published. Modelling code is available on the github: https://github.com/statbiophys/zebrafishGC, (copy archived at *Statistical Biophysics Consortium, 2023*). Data files are available at: https://doi.org/10.5281/zenodo.6774389.

The following previously published dataset was used:

| Author(s) | Year | Dataset title | Dataset URL | Database and Identifier |
| --- | --- | --- | --- | --- |
| De Vico Fallani F, Corazzol M, Sternberg JR, Wyart C, Chavez M | 2022 | Data access for figures of Chen, Ginoux, Wyart, Mora & Walczak | https://doi.org/10.5281/zenodo.6774389 | Zenodo, 10.5281/zenodo.6774389 |

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
