## [Editor Report]

This important paper provides an in-depth analysis of the advantages and potential pitfalls of the application of Granger Causality (GC) to calcium imaging data, especially regarding various types of pre-processing. The authors' approach uses compelling rigor in arguing their points, and it is very clear how one would go about replicating their work. These results should be of interest to any researcher attempting to analyze calcium imaging data.

---

## [Decision Letter]

**Decision letter after peer review:**

Thank you for submitting your article "Granger causality analysis for calcium transients in neuronal networks: challenges and improvements" for consideration by *eLife*. Your article has been reviewed by 3 peer reviewers, one of whom is a member of our Board of Reviewing Editors, and the evaluation has been overseen by Ronald Calabrese as the Senior Editor. The reviewers have opted to remain anonymous.

Essential revisions:

1) It was not clear to the reviewers what lessons are specific to the system they are studying and which ones are to be taken as more general lessons. Certainly, dealing with slow calcium dynamics, motion artifacts, and smoothing, are general problems in calcium imaging, but the reviewers were puzzled a bit about how to decide which neurons are "strange" without a lot of system-specific knowledge. This seems to be a rather important effect, and having a bit more guidance as to this point in the discussion would be useful.

2) It was not entirely sure what results one should take home from the hindbrain analysis. It is clear that there is a more-or-less global signal modulating all neural activity, but this is a common occurrence in population recordings (often, one subtracts this off via PCA or another means before proceeding). Is the general lack of causal links (via the MVGC at least) a generic phenomenon in recurrent networks, or is there something more system-specific here? Accordingly, it might be interesting to run a recurrent neural network simulation with similar properties to the hindbrain (and perhaps with correlated driving) to see what GC/MVGC would predict. Is there any hope of these methods finding information flow in recurrent networks, or should we restrict the method to networks where we expect the primary mode of information transmission to be feedforward? Either performing simulations or limiting the stated results to feedforward networks would be important.

3) The reviewers pointed out several studies that adapt the classical GC for both electrophysiology data and calcium imaging data (e.g., [1] A. Sheikhattar et al., "Extracting Neuronal Functional Network Dynamics via Adaptive Granger Causality Analysis", PNAS, Vol. 115, No. 17, E3869-E3878, 2018. [2] N. A. Francis et al., "Small Networks Encode Decision-Making in Primary Auditory Cortex", Neuron, Vol. 97, No. 4, 2018. [3] N. A. Francis et al., "Sequential Transmission of Task-Relevant Information in Cortical Neuronal Networks", Cell Reports, Vol. 39, No. 9, 110878, 2022.). In reference [1], a variation of GC based on GLM log-likelihoods is proposed that addresses the issues of non-linearity, non-stationarity, and non-Gaussianity of electrophysiology data. In [2] and [3], a variation of GC using sparse multi-variate models is introduced with application to calcium imaging data. In particular, all three references use the sparse estimation of the MVAR parameters in order to mitigate overfitting and also use corrections for multiple comparisons that also reduce the number of spurious links. We suggest discussing these relevant references in the introduction (paragraphs 2 and 3) and discussion.

4) A major issue of GC applied to calcium imaging data is that the trials are typically limited in duration, which results in overfitting of the MVAR parameters when using least squares (See references [2] and [3] above, for example). The authors mention on page 4 that they use least squares to estimate the parameters. However, for the networks of ~10 neurons considered in this work, stationary trials of a long enough duration are required to estimate the parameters correctly. We suggest that the authors discuss this point and explicitly mention the trial durations and test whether the trial durations suffice for stable estimation of the MVAR parameters (this can be done by repeating some of the results on the synthetic data and using different trial lengths and then assessing the consistency of the detected GC links).

5) The definition of the "knee" of the average GC values as a function of the lag L needs to be a bit more formalized. In Figure 2H using the synthetic data, the "knee" effect is more clear, but in the real data shown in Figure 2I, the knee is not obvious, given that the confidence intervals are quite wide. Is there a way to quantify the "knee" by comparing the average GC values as well as their confidence bounds along the lag axis?

6) Another key element of existing GC methods applied to large-scale networks is dealing with the issue of multiple comparisons: for instance, in Figures 2, 3, 4, 6, 7, and 8, it seems like all arrows corresponding to all possible links are shown, where the colormap indicates the GC value. However, when performing multiple statistical tests, many of these links can be removed by a correction such as the Benjamini-Hochberg procedure. It seems that the authors did not consider any correction of multiple comparisons; we suggest doing so and adding this to your pipeline.

7) In Figure 5C, the values of W_IC for the MV cases seem to be more than 1, whereas by definition they should be less than or equal to 1. Please clarify.

8) Is there evidence that the lateralized and rostrocaudal connectivity of the motoneurons occurs at the time-scale of ~750 ms? Given that this time scale is long enough for multiple synapses, it could be the case that some contralateral and non-rostrocaudal connections could be "real", as they reflect multi-hop synaptic connections. Please clarify.

9) Redundant signals: throughout the brain, it's expected that a population of neurons can encode the same information. It's unclear how GC (both the original and the modified versions) can handle this redundancy. Given how pervasive redundant signals are in the brain, this should be addressed in both simulation and experimental data. For example, in one of the manuscript's simulated networks, replace one neuron with 10 copies of it, each with identical inputs and outputs but with the weights scaled by 1/10. Such a network is functionally equivalent to the original but may pose some challenges for the various versions of GC. This issue may also account for the MVGC results in the hindbrain dataset. It might be more appropriate to apply GC to groups of neurons (as indeed the authors cited), instead of applying it at the single-cell level with redundant signals.

10) Both BVGC and MVGC appear to be extremely sensitive to any outlier signals. The most worrying aspect is that the authors developed their corrections and pipelines with the benefit of knowing the structure of the underlying system, whereas in the case where GC would be most useful, the user would be unable to rely on prior knowledge of the underlying structure. For instance, the motion artifact in Figure 3a-c was a helpful example of a vulnerability of naive GC, but one could easily imagine scenarios involving an unmeasured disturbance (e.g. the table is bumped) causing a similar artifact, but if the experimenter is unaware of such unmeasured disturbances then they will not be included in Z, and hence can result in the detection of widespread spurious links. There is a circularity here that's concerning. It seems that one already needs to have the answer (e.g. circuit connectivity) in order to clean up the data sufficiently for BVGC or MVGC to work effectively. Perhaps the authors would be interested in incorporating ideas from the systems identification literature, which can include the estimation of unmeasured disturbances, perhaps in conjunction with L1 regularization on the GC links. This is certainly out of scope for the present work, but it would be worth acknowledging the difficulties of unmeasured disturbances and deferring a general solution to future work. Similar considerations apply to a common unmeasured neuronal input (e.g. from a brain region not included in the field of view of the imaging).

11) Interpretation – would it be correct to state that BVGC identifies plausible causal links, while MVGC identifies a plausible system-level model? We think these interpretations, carefully stated, might provide a helpful way of thinking about the two GC approaches. Taking the results of the paper together, neither BVGC nor MVGC is definitive – BVGC may overestimate the true number of causal links but MVGC is prone to a winner-take-all phenomenon that may represent just one of many plausible system-level models that can account for the observed data. This should be more clearly stated in the manuscript.

*Reviewer #3 (Recommendations for the authors):*

There are numerous typos in the manuscript and some odd choices in the citations in the introduction. Given that there are still some major conceptual issues to address, I'll defer these to the future.

---

## [Author Response]

Essential revisions:1) It was not clear to the reviewers what lessons are specific to the system they are studying and which ones are to be taken as more general lessons. Certainly, dealing with slow calcium dynamics, motion artifacts, and smoothing, are general problems in calcium imaging, but the reviewers were puzzled a bit about how to decide which neurons are "strange" without a lot of system-specific knowledge. This seems to be a rather important effect, and having a bit more guidance as to this point in the discussion would be useful.

We thank the reviewer for pointing out the lack of quantitative definition of the “strange” neurons. In this revision, we introduce a quantitative criteria to identify atypical calcium traces to exclude from the analysis, by recognizing that calcium transients of neurons decay typically with the same profile, i.e. similar decaying time constant when fitted to an exponential function.

If a calcium trace has a decay constant much longer (> tens of s) than an average trace (~ 0.1-5 s), we infer that it’s not the trace of a neuron but likely a glial cell.

This method can be applied generally to systems with neurons expected to have the same characteristics in terms of excitability and calcium buffer. This is the method we already used to identify the time constant of calcium decay (*τ = 2.5*s) in Figure 3—figure supplement 1 (Figure S7 from first submission) and Results: GC analysis of motoneuron data: slow calcium timescale. In this figure, we now include the overlay of calcium transients from typical neurons as well as from atypical traces (the “strange” neurons). We have also changed the text in Results: GC analysis of motoneuron data: Removing strange neurons as follows:

“In general, cells whose calcium transients showed time decay of tens of seconds were identified as non neuronal cells and excluded from GC analysis (see overlays of calcium transients in Figure 3—Figure Supplement 1)”.

We also included in the discussion that this method can be generally applied to any other systems relying on calcium imaging to infer neuronal activity.

2) It was not entirely sure what results one should take home from the hindbrain analysis. It is clear that there is a more-or-less global signal modulating all neural activity, but this is a common occurrence in population recordings (often, one subtracts this off via PCA or another means before proceeding). Is the general lack of causal links (via the MVGC at least) a generic phenomenon in recurrent networks, or is there something more system-specific here? Accordingly, it might be interesting to run a recurrent neural network simulation with similar properties to the hindbrain (and perhaps with correlated driving) to see what GC/MVGC would predict. Is there any hope of these methods finding information flow in recurrent networks, or should we restrict the method to networks where we expect the primary mode of information transmission to be feedforward? Either performing simulations or limiting the stated results to feedforward networks would be important.

The reviewers are right. We have now replaced the network in Figure 1 with a more realistic network with feedback. We have also added a figure in the Supplementary Information (Figure 1—figure supplement 1), that tests the performance of GC on VAR dynamics run for randomly connected neurons (N = 10). Each underlying true interaction structure is sampled based on, *P*_connect_, the probability that one neuron has a synaptic connection with the other, and also the connection strength *c*. As shown in Figure 1— figure supplement 1, MVGC performs well in identifying the true and only the true links when the interaction strength is large.

3) The reviewers pointed out several studies that adapt the classical GC for both electrophysiology data and calcium imaging data (e.g., [1] A. Sheikhattar et al., "Extracting Neuronal Functional Network Dynamics via Adaptive Granger Causality Analysis", PNAS, Vol. 115, No. 17, E3869-E3878, 2018. [2] N. A. Francis et al., "Small Networks Encode Decision-Making in Primary Auditory Cortex", Neuron, Vol. 97, No. 4, 2018. [3] N. A. Francis et al., "Sequential Transmission of Task-Relevant Information in Cortical Neuronal Networks", Cell Reports, Vol. 39, No. 9, 110878, 2022.). In reference [1], a variation of GC based on GLM log-likelihoods is proposed that addresses the issues of non-linearity, non-stationarity, and non-Gaussianity of electrophysiology data. In [2] and [3], a variation of GC using sparse multi-variate models is introduced with application to calcium imaging data. In particular, all three references use the sparse estimation of the MVAR parameters in order to mitigate overfitting and also use corrections for multiple comparisons that also reduce the number of spurious links. We suggest discussing these relevant references in the introduction (paragraphs 2 and 3) and discussion.

We thank the reviewers for pointing out these references. While our GC pipeline focuses on preprocessing the calcium signal by denoising, and post-processing the resulting GC matrix by using an adaptive threshold, these previous studies mentioned by the reviewers emphasize more on modifying the underlying dynamics for GC in order to address the non-linear and non-gaussian dynamics in interacting neurons. In future work, it is of our interest to combine these two approaches. To bring this understanding to the readers, we have incorporated the following change of text. We have included in the introduction (paragraph 3) the following:

“New variations of GC accounting for sparsity in the connection have also been developed and applied on single-cell level to both spiking and calcium signals in the primary auditory cortex of mice (Sheikhattar et al. 2018; Francis et al., 2018, 2022).”

We also modified the Discussion section to include the above reference and added in the text, after we review the two important steps of our GC pipeline, the paragraph connecting our approach better to the body of literature:

“Our pipeline focuses on the pre-processing of the signals and the post-processing of the GC results. This effort is complementary to previous studies where new variations of GC were introduced to take into account the non-linear and non-gaussian dynamics in interacting neurons, for example by modifying the underlying VAR dynamics with GLM or VAR with sparsity constraints, and subsequently using adaptive thresholds for significance tests (Kim et al., 2011; Sheikhattar et al., 2018; Francis et al., 2018, 2022). Future work of interest includes combining these variations of GC with our pipeline.”

4) A major issue of GC applied to calcium imaging data is that the trials are typically limited in duration, which results in overfitting of the MVAR parameters when using least squares (See references [2] and [3] above, for example). The authors mention on page 4 that they use least squares to estimate the parameters. However, for the networks of ~10 neurons considered in this work, stationary trials of a long enough duration are required to estimate the parameters correctly. We suggest that the authors discuss this point and explicitly mention the trial durations and test whether the trial durations suffice for stable estimation of the MVAR parameters (this can be done by repeating some of the results on the synthetic data and using different trial lengths and then assessing the consistency of the detected GC links).

We thank the reviewers for pointing out the importance of the length of the time series in the GC calculation and the information missing to describe our dataset lengths. We address this comment with three additional pieces of supporting evidence.

First, we perform the simulation suggested by the reviewer. With the same underlying structure, we simulate two trials with length *T*, measure the GC values using each trial, and compute the Pearson’s correlation coefficient between the two sets of GC values as a measure of consistency of detecting GC links. The results are included in the Figure 1—figure supplement 3, showing that the stronger the coupling is, the smaller the sample size is required to reconstruct a consistent map of information flow.

Second, to compare with the ground truth, we use synthetic data to check the error rate of link detection as a function of trial lengths, the result of which is included in Figure 1—figure supplement 2. Specifically, we simulate GLM-Calcium dynamics on randomly connected networks, with a fixed probability of connection and a range of connection strength, for different trial durations, and we compute the accuracy of link detection. For MVGC, the accuracy depends on the connection strength. If the connection strength is large, then the true links are mostly detected. If the connection strength is small, then more true links are missed.

These numerical results prompt us to test how consistent the inferred GC matrix is for the zebrafish data, if we use only half of the data. If half of the existing time points are already sufficient to give consistent GC structures, then using all data (and more data) suffices for stable estimation. This is the case for the motoneuron data, which we show in the scatter plot of GC values inferred using the first halves vs. the second halves of each experimental trial. The GC values from the two halves have a Pearson’s correlation coefficient of 0.75 for MVGC, and 0.87 for BVGC. The results for motor neurons in the embryonic spinal cord are shown in an additional supplementary figure, Figure 2—figure Supplement 2.

For hindbrain data, the consistency is not as good. The Pearson’s correlation coefficient is 0.41 for MVGC and 0.76 for BVGC. We note that the similarity is higher for the motoneuron dataset due to the cyclic activity pattern leading to two similar halves.

We added in *Results: GC analysis of motoneuron data: Naive GC* the above results, specifically addressing the point of trial duration:

“… In order to verify that the length of the motoneuron dataset is sufficient to estimate the functional connectivity using Granger causality (see Figure 1—figure Supplement 2 and Figure 1--—figure supplement 3 for illustrations using synthetic data), we compared the two networks obtained by applying GC analysis to each half of the calcium imaging time series. The Pearson's correlation coefficient between the GC values from the two halves is 0.87 for BVGC, and 0.75 for MVGC (see Figure 2—figure Supplement 2), indicating the trial length is long enough for a consistent estimation of the GC values.”

We also added a supplementary figure to discuss the overfitting issue (Figure 8—figure Supplement 3). Specifically, we compare the resulting GC links for the hindbrain data, using VAR dynamics, and VAR plus regularization. We show that in MVGC, not observing many links is not purely a problem of overfitting. And that if including regularization, one still needs to use the adaptive threshold to increase the connection.

5) The definition of the "knee" of the average GC values as a function of the lag L needs to be a bit more formalized. In Figure 2H using the synthetic data, the "knee" effect is more clear, but in the real data shown in Figure 2I, the knee is not obvious, given that the confidence intervals are quite wide. Is there a way to quantify the "knee" by comparing the average GC values as well as their confidence bounds along the lag axis?

We thank the reviewers for asking for clarification. We can compare the average GC values as suggested, or the individual pairwise GC values when adding an extra lag. Using 3 lags or more yields very similar results, so we took the decision to keep the simplest model with fewest lags. To clarify, we added in Results: GC analysis of motoneuron data: Naive GC:

“This knee corresponds to the maximum lag with the best balance between accuracy and complexity: further increasing the maximum lag yields very similar results but the GC analysis becomes much more computationally expensive (Figure 2—figure supplement 1).”

The additional Figure 2—figure supplement 1 shows the “Comparison of Granger causality analysis results at different maximum lags. Correlation between the GC values obtained for each pair of neurons using max lag *L* and max lag *L + 1*. All fish (n=9) of the motoneuron data set are used and represented by color (same as Figure 7). The similarity in Granger causality values, measured by the Pearson correlation coefficient *r*, increases as *L* increases and then stabilizes after maximum lag *L = 3* (r = 0.97 both for BVGC and MVGC between *L = 3* and *L = 4*, as well as between L = 4 and *L = 5*), justifying our decision to use *L = 3*. Using a larger maximum lag does not bring more information but brings more complexity, and increases the computation time (especially for MVGC).”

6) Another key element of existing GC methods applied to large-scale networks is dealing with the issue of multiple comparisons: for instance, in Figures 2, 3, 4, 6, 7, and 8, it seems like all arrows corresponding to all possible links are shown, where the colormap indicates the GC value. However, when performing multiple statistical tests, many of these links can be removed by a correction such as the Benjamini-Hochberg procedure. It seems that the authors did not consider any correction of multiple comparisons; we suggest doing so and adding this to your pipeline.

In our original analysis, we had performed multiple comparisons by applying a Bonferroni correction to the p-value threshold. We now clarify this in the main text by adding in *Results: Granger Causality*, after Equation 7,

“…with the number of parameters adjusted accordingly in the reduced and the full model. The significant test is adjusted using multiple comparisons by applying a Bonferroni correction to the *p*-value threshold.”

7) In Figure 5C, the values of W_IC for the MV cases seem to be more than 1, whereas by definition they should be less than or equal to 1. Please clarify.

We thank the reviewers for carefully reading our submission. In our first submission, W_IC_ can be more than 1 when the averaged GC value for contralateral links is negative (Equation 10)*.* This negative GC value can happen when the residuals of fitting from the full and from the reduced model are similar, and after dividing by the degrees of freedom, *(T_regr –_ M_r_)* and *(T_regr –_ M_f_)*, respectively, leads to var(ε~)<var(ε). We did not impose that GC value has to be nonnegative in our definition.

To improve clarity, we have introduced in the revisions the non-negativeness in the definition of GC in Equation 4 and 5 such that GC is defined as the maximum between 0 and the log of ratio of residuals. This way, the *W_IC_* and *W_RC_* will always be less than or equal to 1. We have changed Figure 5C accordingly. The problem of negative GC value does not occur in the results of GC applied to real motoneuron and hindbrain data.

8) Is there evidence that the lateralized and rostrocaudal connectivity of the motoneurons occurs at the time-scale of ~750 ms? Given that this time scale is long enough for multiple synapses, it could be the case that some contralateral and non-rostrocaudal connections could be "real", as they reflect multi-hop synaptic connections. Please clarify.

In the spinal cord, motor neurons form synapses onto muscle fibers and are not synaptically connected among themselves. At the embryonic stage, motor neurons are part of an electronically-coupled network relying on GAP junctions. During twitches, motor neurons are recruited by excitatory neurons located in the hindbrain and enable their sequential activation from head to tail. We therefore use this dataset to infer flow of information, not synaptic connectivity. This important point is now added to the result section.

Physiologically, the two chains of motor neurons are not directly connected to each other. However, all motor neurons are controlled on each side by common driver neurons, in order to be sequentially activated (St Amant and Drapeau, Neuron 2001). Overall, each chain of neurons on one side can be recruited at a frequency up to 2hz, so with a typical 250ms time difference (Warp et al., Current Biology 2012; Wan et al., Cell 2019 https://pubmed.ncbi.nlm.nih.gov/31564455/). Depending on the position of a neuron in the focal plane of the spinning disk microscope, we have good reasons to think that the kinetics of the calcium transient could appear fast (in the plane) versus slow (out of the focal plane). This artifact may lead to variations in the apparent pattern of recruitment of motor neurons in the rostral caudal axis. This could explain why in the data, we sometimes observe differences in the recruitment pattern leading to a caudal motor neuron being activated a few hundreds milliseconds before a rostral one.

This physiological structure is explained in Results: GC analysis for motoneuron data: The data, when the motoneuron dataset is introduced. To further clarify that the ground truth is not strictly *W_RC_ = 1*, we added in Results: GC analysis for motoneuron data: Defining directional biases, after the definition of

*W_RC_*:

“Notice that based on the anatomy, the ground truth of information flow can be smaller than *W_RC_ = 1*, because motoneuron activity has been shown to rely on command neurons in the brainstem that activates motoneurons sequentially in the spinal cord, not that the motor neurons synapse onto each other.”

9) Redundant signals: throughout the brain, it's expected that a population of neurons can encode the same information. It's unclear how GC (both the original and the modified versions) can handle this redundancy. Given how pervasive redundant signals are in the brain, this should be addressed in both simulation and experimental data. For example, in one of the manuscript's simulated networks, replace one neuron with 10 copies of it, each with identical inputs and outputs but with the weights scaled by 1/10. Such a network is functionally equivalent to the original but may pose some challenges for the various versions of GC. This issue may also account for the MVGC results in the hindbrain dataset. It might be more appropriate to apply GC to groups of neurons (as indeed the authors cited), instead of applying it at the single-cell level with redundant signals.

We agree with the reviewer that the problem of redundant signals is subtle. Redundant signals are not equivalent to repeated signals. A repeated signal is an exact copy of another signal, which experimentally could come from a mis-partition of a single neuron. In this case, GC is unable to detect the correct links, especially the connection between the repeated neuron and the others. In contrast, redundant encoding is more in the structure of the network. Because of stochasticity, the dynamics of two neurons – with identical input and output weights to other neurons – can be different, as we test it on the network in Figure 1 with VAR and with GLM-Calcium dynamics. Such a difference in the dynamics allows MVGC to detect the true structure despite redundant encoding. We have added a paragraph in Results: GC analysis of a small synthetic neuronal net, and an additional supplementary figure (Figure 1 —figure supplement 7) to clarify this issue. The added text is:

“Redundant signals. In the brain, a population of neurons can encode redundant signals. If the redundancy is in the structure of the network, for example, when two neurons share the identical input and output, the stochasticity will lead to different neuronal dynamics, and MVGC is able to identify the true connection. However, if two neurons have exactly the same activity, MVGC will underestimate the causality, while BVGC can still reveal these connections (Figure 1— figure Supplement 7).”

The additional Figures 1 —figure supplement 7 shows:

Presence of redundant signals harms the performance of MVGC. GC analysis performed on synthetic data generated using VAR dynamics on the network structure in the main figure, for *T = 5000* data points. GC reveals all links for networks with redundant structures, where an “artificial” neuron 11 is created with identical input and output strength as neuron 1 (*redundant structure, middle column*). However, MVGC fails to identify causal links when the signal from neuron 1 is copied to create another “artificial” neuron 11 (*redundant signal, right most column*). BVGC is able to identify the underlying true connectivity.

For the hindbrain data in the last figure, we previously tried grouping the neurons. However, we find that the most correlated neurons are spatially located far from each other, as one would expect given that pairs of pre- and postsynaptic neurons may be connected through a very long axon. This makes grouping difficult: grouping neurons by correlation will disable us finding the spatial information flow, while grouping neurons by spatial proximity washes away meaningful signals.

10) Both BVGC and MVGC appear to be extremely sensitive to any outlier signals. The most worrying aspect is that the authors developed their corrections and pipelines with the benefit of knowing the structure of the underlying system, whereas in the case where GC would be most useful, the user would be unable to rely on prior knowledge of the underlying structure. For instance, the motion artifact in Figure 3a-c was a helpful example of a vulnerability of naive GC, but one could easily imagine scenarios involving an unmeasured disturbance (e.g. the table is bumped) causing a similar artifact, but if the experimenter is unaware of such unmeasured disturbances then they will not be included in Z, and hence can result in the detection of widespread spurious links. There is a circularity here that's concerning. It seems that one already needs to have the answer (e.g. circuit connectivity) in order to clean up the data sufficiently for BVGC or MVGC to work effectively. Perhaps the authors would be interested in incorporating ideas from the systems identification literature, which can include the estimation of unmeasured disturbances, perhaps in conjunction with L1 regularization on the GC links. This is certainly out of scope for the present work, but it would be worth acknowledging the difficulties of unmeasured disturbances and deferring a general solution to future work. Similar considerations apply to a common unmeasured neuronal input (e.g. from a brain region not included in the field of view of the imaging).

We corrected for unmeasured inputs by preprocessing the data using characteristic features for calcium transients, and not assuming any underlying network structures. As we expect, all calcium transients follow a fast rise and a slow decay that can be fitted by exponentials, disturbances such as the table being bumped can be identified and corrected. We also emphasize that knowing the underlying structure of the motoneuron network only prompts us to clean up the correlated noise, and that this method developed is applicable to general datasets without a priori known structures, such as the hindbrain dataset.

We thank the reviewers for pointing out literature in system identification theory. To provide more detailed context, we added in the Discussion section sentences on recent development of GC and other causality measures that considers hidden input, and discussed their advantages and disadvantages:

“Firstly, we emphasize that with current experimental techniques, only a small portion of the entire nervous system is recorded. While newer versions of GC have been adapted to take into consideration latent variables which contribute equally to all neurons [49], or through subsequent removal of causal links [50], GC does not in general allow identification of the true underlying connection in the presence of unobserved neurons. However, it is able to identify information flow and an effective functional causal network for the observed neurons.”

11) Interpretation – would it be correct to state that BVGC identifies plausible causal links, while MVGC identifies a plausible system-level model? We think these interpretations, carefully stated, might provide a helpful way of thinking about the two GC approaches. Taking the results of the paper together, neither BVGC nor MVGC is definitive – BVGC may overestimate the true number of causal links but MVGC is prone to a winner-take-all phenomenon that may represent just one of many plausible system-level models that can account for the observed data. This should be more clearly stated in the manuscript.

We agree with the referees, and have added the following sentences to *Discussion*:

“We also compared results from bivariate GC and multivariate GC analysis. While BVGC overestimate the true number of causal links, MVGC is prone to a winner-take-all phenomenon that may represent just one of many plausible system-level models that can account for the observed data. This is especially problematic when the signals are strongly correlated, for example, in the presence of redundant signals. In addition, MVGC is more prone to the issue of overfitting: we suggest using BVGC on datasets with large numbers (hundreds) of neurons, and MVGC on datasets with few tens of neurons. In general, starting with a small number of neurons of interest seems wise.”